# Depth-resolved particle associated microbial respiration in the northeast Atlantic

A. Belcher[1,2], M. Iversen[3,4], S. Giering[1], V.Riou[5], S.A. Henson[1], L, Berline[5], L.Guilloux[5], R. Sanders[1]

[1] National Oceanography Centre, Southampton, SO14 3ZH, UK
[2] University of Southampton, Southampton, SO14 3ZH, UK
[3] Helmholtz Young Investigator Group SEAPUMP, Alfred Wegener Institute for Polar and Marine Research, Bremerhaven, Germany
[4] MARUM, University of Bremen, Bremen, Germany
[5] Aix-Marseille Université, Mediterranean Institute of Oceanography (MIO), UM 110 CNRS/INSU, IRD, 13288 Marseille; Université du Sud Toulon-Var, 83957 La Garde, France

*Correspondence to*: A. Belcher (A.Belcher@noc.soton.ac.uk)

**Abstract.** Atmospheric levels of carbon dioxide are tightly linked to the depth at which sinking particulate organic carbon (POC) is remineralized in the ocean. Rapid attenuation of downward POC flux typically occurs in the upper mesopelagic (top few hundred meters of the water column), with much slower loss rates deeper in the ocean. Currently we lack understanding of the processes that drive POC attenuation, resulting in large uncertainties in the mesopelagic carbon budget. Attempts to balance the POC supply to depth discrete layers of the mesopelagic with respiration by zooplankton and microbes in those layers rarely succeed. In particular, it has been suggested that organic-carbon supply exceeds respiration by free-living microbes and zooplankton in the upper mesopelagic. We test the hypothesis that particle-attached microbes contribute significantly to community respiration in the mesopelagic, measuring particle associated microbial respiration of POC, through shipboard measurements on individual marine snow aggregates collected at depth. We find very low rates of both absolute and carbon-specific particle associated microbial respiration ($<3\%$ $d^{-1}$), suggesting that this term cannot solve imbalances in the upper mesopelagic POC budget. The relative importance of particle associated microbial respiration increases with depth, accounting for up to 33% of POC loss in the mid mesopelagic (128-500 m). We suggest that POC attenuation in the upper mesopelagic is driven by the transformation of large, fast sinking particles to smaller, slowly sinking and suspended particles via processes such as zooplankton fragmentation, and that this shift to non-sinking POC may help to explain imbalances in the mesopelagic carbon budget.

## 1 Introduction

The biological carbon pump plays a key role in regulating the partitioning of carbon dioxide ($CO_2$) between the ocean and atmosphere, and without it atmospheric $CO_2$ would likely be 200 ppm higher than it is today (Parekh et al., 2006). Key

to determining its effectiveness is the efficiency with which organic carbon sinks through the ocean interior (quantified as the transfer efficiency), and thus the depth at which material is remineralized (Kwon et al., 2009). However, despite its importance, the processes governing the loss of organic carbon within the mesopelagic are poorly understood (Burd et al., 2010).

5      POC sinking out of the euphotic zone can be transformed within the mesopelagic in many ways including, zooplankton feeding, fragmentation via sloppy feeding, microbial solubilization to dissolved organic carbon (DOC), and physically driven aggregation and disaggregation processes (e.g. Azam and Malfatti, 2007; Burd and Jackson, 2009; Belcher et al., 2016). Ultimately carbon is lost from the organic carbon pool as $CO_2$ via respiration, and hence, in theory at steady state, the loss (attenuation) of POC should be balanced by community respiration (Buesseler and Boyd, 2009). However, 10  settling organic matter is often found to be insufficient to meet the energy demands of microbes in the dark ocean, thus leading to an imbalanced mesopelagic carbon budget (Herndl and Reinthaler, 2013; Steinberg et al., 2008).

A recent study managed to close the mesopelagic carbon budget between 50-1000 m in the North Atlantic (Giering et al., 2014). However, this study contained large and compensating imbalances between sources and sinks in upper and lower mesopelagic layers, with an excess of POC supply to the upper mesopelagic (50-150 m depth), and an excess of respiration 15  in the lower mesopelagic (150-1000 m depth). Prokaryotes were found to be responsible for most of the respiration (92%) across both depth ranges, however the sampling techniques used may have underestimated the respiratory loss due to particle associated prokaryotes which have typically not been included in mesopelagic carbon budget studies (Giering et al., 2014; Steinberg et al., 2008). Data from the subtropical North Atlantic and west Antarctic Peninsula show that particle associated microbial respiration can contribute 32-93% of the total respiration measured in situ (McDonnell et al., 2015), suggesting 20  that particle associated microbes could play an important role in the loss of POC in the mesopelagic. Alternatively, Iversen et al., (2010) found that rapid flux attenuation was best explained by flux feeding by zooplankton off Cape Blanc (Mauritania). We hypothesize that POC losses via particle associated respiration (a term not directly measured by Giering et al., 2014 or Steinberg et al., 2008) may help to address imbalances in the upper mesopelagic carbon budget.

Marine snow particles (aggregates of detritus, living organisms and inorganic matter larger than 0.5 mm in diameter, 25  Alldredge and Silver, 1988) can make up a large fraction of the sinking POC in the ocean and host microbial abundances 2-5 orders of magnitude higher than those found free-living in the surrounding water column (Silver and Alldredge, 1981; Thiele et al., 2015). The fragile nature of marine snow particles makes sampling and measurement difficult; many previous measures of particle associated respiration have been carried out on roller tank formed marine snow aggregates, either from lab cultures of phytoplankton or natural sea water samples (Grossart and Ploug, 2001; Iversen and Ploug, 2010, 2013; 30  Iversen et al., 2010). A few experiments have utilized SCUBA or submersibles to collect in situ aggregates and estimated heterotrophic bacterial production by measuring leucine uptake (Alldredge and Youngbluth, 1985; Smith et al., 1992) with few measuring respiration directly on individual aggregates (Ploug et al., 1999). To the best of our knowledge only two studies have combined direct measures of respiration on aggregates collected at depth with measurements of POC flux (Collins et al., 2015; McDonnell et al., 2015), both of which lack sufficient vertical resolution in the upper mesopelagic to

capture the region of most rapid change. Collins et al. (2015) measure rates of substrate-specific microbial respiration of $0.007\pm0.003$ $d^{-1}$ to $0.173\pm0.105$ $d^{-1}$ in the North Atlantic, with rates of $0.01\pm0.02$ $d^{-1}$ and $0.4\pm0.1$ $d^{-1}$ measured by McDonnell et al. (2015) at the Western Antarctic Peninsula and Bermuda Atlantic Time Series respectively. Previous studies are therefore inconclusive as to the importance of particle associated microbes on the attenuation of POC, with some studies suggesting they play a minor role (Alldredge and Youngbluth, 1985; Collins et al., 2015; Ducklow et al., 1982; Karl et al., 1988) and others suggesting a larger contribution (Iversen and Ploug, 2013; Ploug et al., 1999; Turley and Stutt, 2000).

To build upon these previous studies we assess the role of particle associated microbial respiration in POC flux attenuation, presenting a vertical profile of particle associated respiration rates measured on individual marine snow particles collected at depth. In an attempt to assess whether this term can improve our ability to balance the fast sinking POC budget in the upper mesopelagic, we make these measurements in the northeast Atlantic at the site of Giering et al., (2014) where we have the most complete knowledge of the mesopelagic carbon budget. In addition, we compare the natural particle sinking velocities and respiration rates with those of aggregates produced in roller tanks. We focus on the upper region of the mesopelagic (mixed layer depth-500 m) where the most rapid attenuation occurs, a region that is poorly understood and poorly represented in model studies (Henderson and Marchal, 2015).

## 2 Methods

### 2.1 Study site

Measurements were made during research cruise DY032 ($20^{th}$ June – $8^{th}$ July 2015) to the Porcupine Abyssal Plain (PAP) observatory site (49 °N, 16.5 °W) in the northeast Atlantic aboard RRS *Discovery*. Vertical profiles of the water column at each site were made using a Conductivity-Temperature-Depth (CTD) unit (Seabird 9Plus with SBE32 carousel). The mixed layer depth (MLD) was determined as the depth where temperature was 0.5°C lower than surface temperature (Monterey and Levitus, 1997).

### 2.2 Chlorophyll-a

Depth profiles of chlorophyll-a were measured at a number of points during the cruise using water samples (200 mL) collected with the CTD rosette. Samples were filtered onto 0.8 μm MPF300 glass fibre filters, and frozen at -20 °C. Pigments were extracted in 90% acetone for 22-24 hours at 4 °C and fluorescence measured on a Trilogy Turner Designs 7200 lab fluorometer calibrated with a pure chlorophyll-a standard (Sigma, UK).

Aqua MODIS 9 km, 8 day satellite chlorophyll-a data (downloaded from the NASA Ocean Biology website; http://oceancolor.gsfc.nasa.gov/cms/) were used to assess mesoscale variability (e.g. passage of eddies) during the sampling period. We examine changes in surface chlorophyll prior to and post sampling by averaging chlorophyll data over the study region (48.5-49.5 °N, 16.0-17.0 °W).

## 2.3 Particle flux and composition

Particle flux and composition were measured using Marine Snow Catchers (MSC), large (95 L) PVC closing water bottles designed to minimize turbulence (Riley et al., 2012). MSCs were deployed between 36-500 m during the course of the cruise, at which depth the bottles were closed, retrieved on deck and left for particles to settle. Deployment depths were chosen based on MLD determined as the depth with steepest gradient in temperature from the most recent CTD profile. MSC deployments were carried out during the day with the exception of the two samples at 36 m and 128 m, which were deployed at night due to logistical limitations. Particles were allowed to settle onto a particle collector tray at the base of the MSC for two hours (we deem these 'fast sinking' as in Riley et al., (2012)), after which those visible by eye (>0.15 mm diameter) were picked from three quadrants using a wide bore pipette, filtered onto pre-combusted (450 °C, 24 h) glass fibre filters (25 mm diameter GF/F, Whatman), and oven dried at 50 °C for replicate analysis of POC. Filters were subsequently fumed with 37% HCl in a vacuum desiccator for 24 hours, and dried for 24 hours at 50 °C. Filters and filter blanks were placed in pre-combusted (450 °C, 24 h) tin capsules as in Hilton et al., (1986), and POC measured by a CE-440 Elemental analyser (Exeter Analytical.285 Inc). Particles in the remaining quadrant were transferred to a temperature controlled laboratory (10 °C) and used for measurements of sinking and respiration rates (section 2.4). Slow sinking particles were also collected as of Riley et al., (2012), with slow sinking particle velocities calculated using the SETCOL method (Bienfang, 1981). Slow sinking fluxes were only a small part of the total sinking flux (on average <10%) and due to their slower sinking rate these particles do not penetrate as deeply in the mesopelagic. Hence we focus our study on fast sinking particles only.

The flux of POC ($F$ in mg C m$^{-2}$ d$^{-1}$) associated with fast sinking particles was calculated as follows:

$$F = \frac{m}{A} \times \frac{w}{h},\tag{1}$$

where $m$ refers to the total mass (mg) of fast sinking POC collected from the MSC, $A$ the area (m$^2$) of the MSC based on inner MSC diameter, $w$ the measured sinking velocity (m d$^{-1}$) from laboratory measurements, and $h$ the height of the snow catcher (1.53 m). Sinking velocities of marine aggregates were measured in a flow chamber (section 2.4), and the median value for each depth horizon used to avoid bias by rare fast sinking aggregates. The rate of particle flux attenuation was assessed by fitting a power-law function (Martin et al., 1987) to the flux data,

$$F_z = F_0 \times (z/z_0)^{-b},\tag{2}$$

where $z$ is the depth of the flux, and $F_0$ is flux at the reference depth (in this case 26 m, i.e. the mixed layer depth). A high value of $b$ corresponds to high attenuation (shallow remineralization) and vice versa. We note that as in situ particle production at depth is not considered, this represents a lower bound estimate of flux attenuation.

The type of fast sinking particles at each depth was assessed under a microscope and photographs taken using a Leica DM-IRB inverted microscope and Canon EOS 1100D camera. Particles were classified into phytodetrital aggregates (aggregations >0.15 mm equivalent spherical diameter (ESD) containing phytoplankton cells and other phytodetrital material, herein referred to as PA), faecal pellets (FP), and unidentified phytodetritus. Individual particle dimensions were measured using ImageJ (version 1.49p) and volumes calculated using formulae for a sphere, prolate ellipsoid or cylinder

depending on particle shape. Conversions from PA volume were based on measurements of POC content of marine aggregates collected at depth (section 2.4), and a carbon to volume ratio of 0.08 mg mm$^{-3}$ used for FP based on literature estimates (range 0.01-0.15 mg mm$^{-3}$) (Wilson et al., 2008). FP carbon content can vary greatly even within species depending on factors such as food type and concentration (Urban-Rich, 2001), which introduces uncertainty into our estimates of their contribution to the total POC flux.

## 2.4 Oxygen gradients in marine snow aggregates

The rates at which sinking particles were degraded due to the respiration of particle associated microbes were calculated from direct measurements of oxygen gradients within PA. PA were transferred into a temperature controlled flow chamber system (Ploug and Jorgensen, 1999) containing filtered sea water (0.22 µm), taken from the MSC deployed at 36 m and maintained at 10 °C (at the low end of temperatures measured during the study, Fig. 1). Only one incubation temperature was possible due to laboratory and space limitations. The salinity in the flow chamber was 35.5 PSU which, considering the low variation in salinity profiles (standard deviation of 0.008 PSU at 36 m depth) should represent conditions at all depths sampled. Within 24 hours of collection, PA were placed carefully in the flow chamber using a wide bore pipette. The wide bore pipette lifts the particles with the surrounding water so that the particles remain suspended in water during the handling and minimal physical stress is exerted on the particles. The microbial communities associated with the aggregates are not removed by this method (Kiørboe et al., 2002).The x, y, and z dimensions of PA were measured using a horizontal dissection microscope with a calibrated ocular, and three measurements of the sinking velocity made for each PA by suspending the PA with an upward flow (Ploug et al., 2010). PA volumes were calculated from x, y, z dimensions based on an ellipsoid, and equivalent spherical diameters (ESD) calculated.

A profile of oxygen was measured from the ambient water, through the diffusive boundary layer (DBL) and into the PA using a Clark-type oxygen microelectrode and guard cathode (Revsbech, 1989) mounted in a micromanipulator. Measurements were made in increments of (50-200 $\mu$m) on the downstream side of the particle and oxygen fluxes calculated using a diffusion-reaction model based on Fick's first law of diffusion (diffusion coefficients of 1.4691 x10$^{-5}$ cm$^2$ s$^{-1}$ for 10 °C and salinity 35 PSU, Broecker and Peng, 1974). Two to three replicate profiles were taken for each PA where possible. We used a solver routine to find the optimum solution minimizing the sum of the squares between measured and modelled oxygen concentrations (see Ploug et al., (1997) for full details). Total oxygen consumption within the PA was calculated using the equation for the surface area of an ellipsoid assuming that net oxygen fluxes do not vary significantly on the upstream and downstream sides (Ploug and Jorgensen, 1999). As oxygen consumption in the DBL is a measure of the respiration rate of the microbial community associated with the PA due to net exchange of oxygen via molecular diffusion, the carbon respiration ($C_{resp}$ in mg C mm$^{-3}$ d$^{-1}$) can be calculated based on a respiratory quotient (RQ) of 1 mol $O_2$ to 1 mol $CO_2$ (Ploug and Grossart, 2000; Ploug et al., 1997). This was chosen as a conservative value in the range of literature values typically applied (0.7-1.2 mol:mol) for respiration of carbohydrates and lipids (Berggren et al., 2012), but adds uncertainty to

our estimates that cannot be better constrained without knowledge of the form of carbon within the PA utilized for microbial respiration.

Following respiration measurements, PA were stored in 1.5 mL Eppendorf tubes before pooling PA into size classes based on ESD and placing onto pre-combusted (450 °C, 24 h) glass fibre filters (25 mm diameter GF/F, Whatman) for measurement of POC as described in section 2.3. This enabled the carbon content per unit volume (mg C mm$^{-3}$) for each size class at each depth range to be calculated and hence POC content of individual PA to be estimated. We measured POC to volume ratios of two sizes classes (typically <0.6 mm ESD, >0.6 mm ESD) at each depth horizon (with the exception of samples at 128, 200 and 500m where all measured particles were <0.6 mm ESD so only one size class was used), to take into account the fractal geometry of aggregates and non-linear volume to POC ratio (Alldredge, 1998).These PA POC contents ([POC] in mg C mm$^{-3}$) were then used to calculate carbon-specific respiration rates ($C_{spec}$ in d$^{-1}$) as follows:

$$C_{spec} = C_{resp}/[POC].$$
(3)

## 2.5 Roller tank experiments

Water was sampled on 24[th] June from 12 m depth (within the surface peak in chlorophyll) from the CTD rosette and transferred to 2 L acid-cleaned Nalgene polycarbonate bottles. The bottles (141 mm diameter, 249 mm height) were rotated on a 120V Benchtop Roller Culture Apparatus (Wheaton) at 3 rotations per minute (rpm, Iversen and Ploug, 2010) at 10 °C in the dark. The tanks were left to incubate in the dark and form aggregates for a period of 9 days before carefully removing aggregates and measuring respiration rates as described above.

## 2.6 Statistics and error analysis

Attenuation of fast sinking POC flux with depth was best described by a power-law relationship fit ($R^2$=0.42, $p$=0.06, n=9) (Martin et al., 1987) compared to an exponential fit ($R^2$=0.30, $p$=0.128, n=9) (of form, $F_z = F_0 \exp(z-z_0/z*)$, as in Buesseler and Boyd (2009)). We also tested for any statistical relationship between carbon-specific respiration rates and depth. All statistics were carried out in RStudio (version 0.98.1091; R development core team, 2014). We calculate a relation between particle ESD and sinking velocity by applying a power law fit to the data using the NLS function in RStudio. The choice of a power law relationship is based on the findings of previous studies (e.g. Iversen and Ploug, (2010)) and the divergence of marine snow aggregates from Stoke's Law due to their fractal rather than spherical geometry.

Time and methodological constraints of measuring very small particles, prohibited us from measuring the respiration rate of every particle collected in the MSC. Hence, before using these measurements to assess the contribution of particle associated microbial respiration to the mesopelagic carbon budget, we must first define upper and lower bounds to our estimates based on our uncertainties. We conduct a Monte Carlo analysis (with 10,000 iterations) of the individual parameters used in the calculations of carbon-specific respiration and remineralization length scale. We randomly sample (with replacement) our measured volumetric oxygen respiration rates at each depth. For each of these randomly selected particles we use the corresponding sinking velocity and ESD in subsequent calculations of carbon-specific respiration and

remineralization length scale. For the RQ, used to convert $O_2$ to $CO_2$, we define a uniform distribution of possible values over the range of RQ values typically applied in the literature (0.7-1.2, Berggren et al., (2012)). PA were pooled into size classes and could only be measured once for POC content. For each depth, we create a normal distribution of possible POC:volume ratios for each size class, with our measured value as the mean, and standard deviation calculated from the standard deviation of the individual aggregate volumes within a size class. Based on the 10,000 iterations for each of the aforementioned parameters we obtain a range of estimates for the remineralization length scale (via particle associated microbial respiration) at each depth. We then use the mean of these distributions ±standard deviation to put error bounds on our estimates of the POC loss via particle associated microbial respiration.

## 2.7 Zooplankton respiration

Zooplankton were sampled in vertical net hauls (0-200 m) at 1 m s$^{-1}$ speed using a 200 µm mesh size WP2 Net with a 57 cm frame diameter, fitted with filtering cod-ends. Collected organisms were fixed directly after collection with formalin at 5% final concentration for further analyses. In the laboratory, fixed samples were digitized with the ZooScan digital imaging system (Gorsky et al., 2010) to determine the size structure of the community. Each sample was divided into two fractions (<1000 and > 1000 µm) for accurate estimation of rare large organisms in the scanned subsample (Vandromme et al., 2012). Fractions were split using a Motoda splitting box until containing approximately 1000 objects. The resulting samples were poured onto the scanning cell and individual zooplankton were manually separated with a wooden spine, in order to avoid overlapping organisms. Each scanned image was later processed using ZooProcess (Gorsky et al., 2010). Each object in the image was automatically classified into five zooplankton categories (copepods, chaetognatha, appendicularia, other crustaceans, other zooplankton) and three non-living categories (detritus, fibers and out of focus) using Plankton Identifier (http://www.obs-vlfr.fr/~gaspari/Plankton_Identifier/index.php). These automatic results were manually validated. Finally, dry weight (DW) of each zooplankton object was estimated from its area using Lehette and Hernández-León's, (2009) allometric relationships corresponding to the five zooplankton categories. Respiration per individual (µC individual$^{-1}$ h$^{-1}$).was computed from DW using relationship from Ikeda et al., (2001) for copepods and (Ikeda, 1985) for other groups:

$$Zooplankton\ Respiration = exp(a_1 + a_2 \ln(DW) + a_3 T) \times RQ \times 12/22.4 \qquad (4)$$

Here DW is dry weight (mg C individual$^{-1}$), RQ is the respiratory quotient (0.8 mol C/mol $O_2$), T is the average temperature over top 200 m (12.5 °C), 12/22.4 is the molar conversion factor and parameters $a_1$, $a_2$, and $a_3$ were dependent on the type of zooplankton. Total zooplankton respiration (0-200 m) was calculated by summing the respiration values for each individual. Day and night respirations were calculated for 16 and 8 hours respectively based on day length at the study site.

# 3 Results

## 3.1 Hydrography and surface chlorophyll-a

The consistency of vertical temperature profiles suggests little variation in water mass structure during the cruise (Fig. 1b). Temperatures ranged from 15.2 °C at the surface to 10.9 °C at 500 m, with salinity remaining relatively constant with depth (average 35.34-35.56, Fig. 1b). The mixed layer shallowed from 32 m to 26 m (Fig. 1b), with peak chlorophyll just above the MLD at 15-25 m, and decreasing from 1.9 to 1.4 mg $m^{-3}$ during the course of the cruise based on discrete measurements (Fig. 1c). The MLD was typically within ±5 m of the 1% photosynthetically active radiation (PAR) level. Satellite chlorophyll data is consistent with in situ data, declining from 1.8 to 1.2 mg $m^{-3}$ in the PAP region, suggesting sampling was carried out in the 'post peak' phase (Fig.1c).

## 3.2 Particle composition

A total of 10 MSC deployments were made over an 11 day period with particle composition and respiration measurements carried out for 7 deployments (Table 1).

The dominant component of fast sinking particles was PA at all depths sampled (one MSC sample per depth, Fig. 2), accounting for 96 % of sinking POC at 36 m and decreasing to 66 % at 500 m associated with an increasing abundance of FP with depth. The lack of FP observed in our sample at 113 m may be due to the heterogeneous distribution of FP at a particular depth associated with patchy zooplankton distributions. The increase in FP numbers below 100 m could be due to an increase in zooplankton populations with depth, zooplankton diel vertical migration, and/or increased FP loss in the upper mesopelagic due to processes such as fragmentation and coprophagy. Qualitative assessment of FP morphology shows that FP were longer, thinner and darker deeper in the water column, implying a change in zooplankton community composition with depth.

## 3.3 Particle sinking velocities

Sinking velocities of natural PA collected at depth ($PA_n$) ranged from 4-255 m $d^{-1}$ (Fig. 3), reflecting both the range in size of PA measured (0.14-1.09 mm ESD) and the heterogeneous composition of PA. Median sinking velocities showed less variability ranging from 11-34 m $d^{-1}$ (10-32 m $d^{-1}$ and 21-62 m $d^{-1}$ for aggregates <0.6 mm (n=74) and >0.6 mm (n=24) ESD respectively), showing consistency in the composition of the bulk of sinking $PA_n$. There was no significant correlation between $PA_n$ sinking velocity and depth for either size class ($R^2$=0.004, $p$>0.1, n=98). $PA_n$ sinking velocity was significantly ($R^2$=0.17, $p$<0.0001, n=98) correlated with ESD (6 outliers, defined as being outside 2 standard deviations from the mean, were excluded in this relationship). The low $R^2$ shows that there is variability around this relationship, suggesting some heterogeneity in PA composition. PA formed in roller tanks, defined here as $PA_r$, were much larger in size (0.54-3.2 mm ESD), but had lower sinking rates for their size (6-173 m $d^{-1}$) as illustrated by the different power-law fits in Fig 3. We note here that sinking velocity measurements were limited to those particles visible by eye (ESD > 0.15 mm). However, measured

velocities do agree with previous observations for the North Atlantic ocean which range from 0.2-181 m d$^{-1}$ (supplementary table S1 in Collins et al. (2015)). A study in the Southern Ocean by Laurenceau-Cornec et al. (2015) also found size-specific (1.3-3.1 mm ESD) sinking velocities similar to those measured here (50-149 m d$^{-1}$).

### 3.4 Particle flux

Consistent with other studies we see a sharp decline in fast sinking POC concentration (not shown) and fast sinking POC flux with depth (Fig. 4). Rapid attenuation in the upper 128 m was followed by a slower decrease and possibly even an increase in POC flux below 128 m, suggesting that different processes may be controlling POC attenuation in the upper and lower mesopelagic, or that the rate of processes varies with depth. Interestingly we see an increase in flux between 203 and 500 m, which may reflect higher surface production in the days prior to sampling (Fig. 1c) and the time taken for material to reach this depth from the surface. Based on a median sinking rate of 34 m d$^{-1}$ measured at 500 m, material at this depth would have originated at the surface on Julian day 164, 15 days prior to sampling, which corresponds to the peak in surface chlorophyll concentrations (Fig. 1c). This increase in flux is associated with twice as much FP POC at 500 m compared to 203 m, and a 29% increase in aggregate POC. Considering the decrease in resident zooplankton populations with depth (Giering et al., 2014), it seems unlikely that FP production was higher at this depth unless there is a large contribution by diel vertical migrators, and may instead reflect reduced FP loss. However, this scenario could also be due to non steady state conditions with high FP at 500 m reflecting high abundances of FP sinking out from shallower in the water column at a time of previously increased FP production. Excluding this potentially non steady state value at 500 m, we calculate a Martin's $b$ value of 0.71 which is in line with previous studies at this study site (Giering et al., 2014; Riley et al., 2012), but note that this fit is just outside the 5% significance level ($R^2$=0.42, $p$=0.06, n=9). To assess the uncertainty surrounding our calculated $b$ value, we have applied a bootstrap analysis with 100,000 simulations giving a mean $b$ of 0.67 ± 0.34 (standard deviation).

### 3.5 Microbial respiration in phytodetrital aggregates

Using the microelectrode approach we found that oxygen concentrations decreased from the ambient water towards the PA surface, reaching a minimum at the centre of the PA (but remaining well above anoxic conditions in all PA measured) (Fig. 5). Average oxygen fluxes to PA$_n$ did not vary significantly with the depth at which particles were collected, ranging from 11.7-19.1 nmol O$_2$ mm$^{-3}$ d$^{-1}$ (Fig. 6), but variability between PA collected at each depth was large (3.1-43.8 nmol O$_2$ mm$^{-3}$ d$^{-1}$ over the depth range measured, Table 2). Volumetric respiration rates on roller tank formed aggregates (PA$_r$) were smaller (2.5-7.7 nmol O$_2$ mm$^{-3}$ d$^{-1}$) than those of PA$_n$, although the large range in rates for PA$_n$ (4.7-37.7 nmol O$_2$ mm$^{-3}$ d$^{-1}$ for PA >0.6mm ESD) makes direct comparison difficult.

For each depth horizon we measure the POC contents of the PA used in respiration experiments (see section 2.4). We find POC contents of PA$_n$ ranging from 11.7-30.6 μg mm$^{-3}$ (average 15.4 μg mm$^{-3}$) for PA <0.6 mm ESD, and 6.7-11 μg mm$^{-3}$ (average 9.0 μg mm$^{-3}$) for PA >0.6 mm ESD. The POC content of both size classes peaked at 46 m depth, showing a general decline below this (supplementary material, table S1). PA$_r$ had lower POC to volume ratios, 2.4 μg mm$^{-3}$ for PA$_r$ <1

mm and 1-2 mm ESD. POC to volume ratios for $PA_r$ are more comparable to phytoplankton culture aggregates formed in roller tanks, which were also similarly sized (Iversen and Ploug, 2010). Our POC measurements are based on filters containing a relatively low number of aggregates; 9-13 aggregates and 4-6 aggregates for aggregates <0.6mm ESD and >0.6mm ESD, respectively. However, despite the low concentrations of carbon measured, sample POC was significantly higher than POC filter blanks (Welch's t-test, $p<0.001$).

### 3.5 Zooplankton respiration

Four zooplankton net tows were carried out during the cruise alongside MSC deployments, one during the day and three at night. Daytime zooplankton DW over the upper 200 m was 313.4 mg DW $m^{-2}$, with night values ranging from 419.1 to 942.7 mg DW $m^{-2}$. Calculated zooplankton respiration rates ranged from 5.1 to 10.1 mg C $m^{-2}$ $d^{-1}$.

## 4 Discussion

### 4.1 Rate of particle associated microbial respiration

Although rates of respiration per PA volume were found to be relatively uniform with depth, we observed variability within each depth range. This may reflect the heterogeneity of aggregate composition in terms of the availability of labile carbon and/or variation in microbial abundance, composition or activity. It may also simply be a result of the range in aggregate sizes at each depth, with higher respiration per volume in smaller aggregates that have higher POC:volume ratios (due to their fractal nature, Logan and Wilkinson, 1990). Our measured 0.14-1.09 mm ESD natural aggregate POC contents are 1.2-10.1 times higher than defined by the size relationship of Alldredge (1998) based on in situ collected 1-5 mm ESD marine snow of mixed composition, and are at the high end of the range of values measured on roller tank formed phytoplankton culture aggregates (0.9-4.6 mm ESD) by Iversen and Ploug (2010). The POC contents of our >0.6mm ESD PA (6.7-11 µg $mm^{-3}$) and 1-2 mm ESD $PA_r$ (2.4 µg $mm^{-3}$) do however compare well with the study of Laurenceau-Cornec et al. (2015) on aggregates formed in roller tanks from in situ collected phytoplankton assemblages; we calculate aggregate POC of 7.4 µg $mm^{-3}$ and 2.7 µg $mm^{-3}$ for aggregates of 0.6 mm and 1.0 mm ESD respectively based on their regression between aggregate volume and POC content (POC=0.58 · $Volume^{0.35}$). This indicates that the power-law defined by Laurenceau-Cornec et al. (2015) can be applied to our study site.

In order to assess whether size related changes in carbon content of PA is the main cause of variability in volume specific respiration rates, we have calculated the carbon-specific respiration rate ($C_{spec}$) (Fig. 7) based on the POC content of individual aggregates. There is a relatively small range in average $C_{spec}$ (0.011-0.014 $d^{-1}$) for $PA_n$ for each depth horizon. Iversen and Ploug, (2010) measured higher rates of $C_{spec}$ (0.13 $d^{-1}$) in roller tank formed phytoplankton culture aggregates of lower POC contents, suggesting that POC content was not the limiting factor for respiration in our study. Despite the consistency of average values of $C_{spec}$ at each depth horizon, there was large variability in $C_{spec}$ for individual aggregates within each depth horizon (full dataset range: 0.002-0.031 $d^{-1}$, Fig, 7). The calculated standard error was consistent across all

depths (0.002—0.003) which implies that the factors driving the variability in $C_{spec}$ are unchanging with depth, suggesting controls on degradation are already determined at shallow depths. Our study does not however account for any changes in respiration that may occur as a result of pressure changes with depth (see section 4.5). If microbes largely attach to particles in the surface ocean (Thiele et al., 2015), the starting abundance of microbes will be in part limited by the residence time of

the particle in the surface ocean as dictated by sinking rate. The highest volume-specific abundances of microbes have been measured on the smallest aggregates (Grossart et al., 2003) which we would expect to have lower sinking velocities. Variable microbial densities, driven by differences in sinking velocity and colonization time, may therefore account for some of the variability in the rate of respiration per aggregate volume or POC content. However, large aggregates could also have high microbial densities following the aggregation of smaller aggregates. There are a number of factors which influence

colonization, and grazing has been modelled to have a higher impact than sinking rate (Kiørboe, 2003).

We must consider that all respiration measurements in this study were carried out at 10 °C (which is just below the temperature measured at 500 m depth), and therefore may not reveal the true vertical structure of particle associated microbial respiration due to the influence of temperature on metabolic rates. To account for this we have applied a Q10 factor of 3.5 from a study on PA (Iversen and Ploug, 2013) and adjusted each $C_{spec}$ to the in situ temperature (dashed line

Fig. 7) with roller tank aggregates being adjusted to the temperature at 12 m (the depth that water was collected from for their formation). In this way we calculate the rate we would expect to be occurring at in situ temperature. This gives higher rates in the upper ocean where temperature changes are higher, but the range with depth is still relatively narrow (average 0.013-0.023 d$^{-1}$, full range 0.002-0.037 d$^{-1}$) and we observe no relationship between PA size and $C_{spec}$ (Fig. 8). In comparison Ploug and Grossart (2000) measured $C_{spec}$ of 0.083 +/- 0.034 d$^{-1}$ on aggregates formed from phytoplankton cultures at 16 °C

which is more comparable to rates of 0.055 +/- 0.006 d$^{-1}$ measured on our roller tank formed aggregates. Iversen and Ploug (2010) measured average $C_{spec}$ of 0.13 d$^{-1}$ at 15 °C but a range of 0.005-0.422 d$^{-1}$ for lab formed aggregates of three different phytoplankton cultures. Similarly, rates of 0.13 d$^{-1}$ (range 0.02-0.36 d$^{-1}$) were measured at 18 °C in aggregates formed in roller tanks from peak fluorescence waters off Cape Blanc, Africa (Iversen et al., 2010). These studies find a lack of size dependency in $C_{spec}$, consistent with our observations. Our measurements are towards the low end of these measurements

which we cannot explain by differences in temperature alone based on a Q10 factor of 3.5 (Iversen and Ploug, 2013). Recalculating the average $C_{spec}$ at each depth based on the upper bound of respiratory quotients that are typically applied in the literature (0.7-1.2, Berggren et al., (2012)), increases our values of 0.019-0.033 d$^{-1}$  which is still lower than the aforementioned studies.

There have been limited measurements made on natural aggregates formed in situ. McDonnell et al. (2015) utilized in

situ incubators to measure $C_{spec}$ of 0.4 d$^{-1}$ at the Bermuda Atlantic Time Series (BATS) station, and 0.01 d$^{-1}$ off the Western Antarctic Peninsula (WAP). Collins et al. (2015) carried out incubations with and without sinking particles collected in the North Atlantic, revealing $C_{spec}$ of 0.007-0.084 d$^{-1}$ with one higher value at 0.173 d$^{-1}$. These rates are more in line with those measured here, yet there are still considerable differences between studies. We turn to a comparison of natural and roller tank formed aggregates in an attempt to explain some of this variability.

## 4.2 In situ versus roller tank formed aggregates

The difficulty of sampling intact PA due to their fragility has led many studies to use roller tanks to create artificial aggregates (e.g. Grossart and Ploug, 2001; Iversen and Ploug, 2010, 2013; Iversen et al., 2010). Even considering the variability in our respiration estimates for $PA_n$, we find much lower respiration rates in $PA_r$ (Fig. 6). This may be in part due to the lower POC to volume ratios (2.4 µg mm$^{-3}$ for $PA_r$ <1 mm and 1-2 mm ESD) which could be a result of POC loss via respiration during incubation in roller tanks. Considering a carbon respiration rate of ~1 µg C mm$^{-3}$ d$^{-1}$ (Fig. 6) and a starting POC content of 9 µg C mm$^{-3}$ d$^{-1}$ (based on average values for $PA_n$), $PA_r$ POC contents could be reduced to 2 µg C mm$^{-3}$ over 7 days (time incubated after first signs of aggregate formation). However, the fractal nature of aggregates means that we would also expect large aggregates to have lower POC to volume ratios (Alldredge, 1998; Logan and Wilkinson, 1990). POC to volume ratios for $PA_r$ are more comparable to those of Iversen and Ploug (2010) which were also similarly sized. To remove the influence of aggregate size, we now compare $PA_n$ and $PA_r$ between 0.7-1.1 mm ESD as this is the size category encompassing both PA types. Volumetric respiration rates of $PA_r$ are 45% lower than $PA_n$, but as POC:volume ratios are 72% lower, this results in $PA_r$ $C_{spec}$ rates that are actually higher than those of $PA_n$ (Fig. 7). This could be due to greater abundances of microbes on individual $PA_r$, and/or higher quality POC that is more readily respired. Although the length of the roller tank incubation may have allowed for a decrease in PA POC via microbial remineralisation, previous studies carrying out longer incubations (Iversen and Ploug, 2013; Iversen and Robert, 2015) did not measure significant changes of $C_{spec}$ over time, suggesting that the long incubation would not bias rates of $C_{spec}$.

The lability of the carbon may differ between naturally formed and roller tank formed aggregates, with $PA_n$ collected at depth containing more reworked material such as FP fragments. Higher rates of solubilization and respiration of labile POC in the euphotic zone would leave more refractory POC to be slowly respired through the mesopelagic and could explain the differences observed between $PA_r$ and $PA_n$. However, we could not visually distinguish any clear differences in taxonomic composition of $PA_n$ collected at depth and roller tank PA based on SEM or light microscope imagery (supplementary Fig. S1). Future studies incorporating analysis of pigments, amino acids and neutral aldoses to determine aggregate age, source material and lability (Cowie and Hedges, 1994; Goutx et al., 2007; Skoog et al., 2008; Tamburini et al., 2009) would help to quantify these differences between $PA_r$ and $PA_n$.

If we consider the process of aggregate formation, microbial populations in roller tanks have a much longer time to attach to and colonize particles due to the infinite residence time created by the rotating bottles. This would allow much higher densities of microbes to colonize the $PA_r$ compared to natural aggregates collected at depth, which, assuming microbial respiration is not limited by any other factor, would drive higher $C_{spec}$. This hypothesis is also consistent with reduced variability in $C_{spec}$ of $PA_r$ compared to $PA_n$ where microbial colonization would be influenced by heterogeneity in particle sinking speed and residence time in the surface ocean. We believe artificially high microbial densities on roller tank formed PA is the most likely cause of differences between our respiration measurements on natural PA collected at depth and measurements on roller tank formed PA. Unfortunately we were not able to to measure bacterial abundance in this study.

Over the past couple of decades there have been numerous studies utilizing roller tanks to create artificial aggregates from natural phytoplankton assemblages in an attempt to replicate natural sinking particulates, investigating processes such as aggregation, sinking velocity, ballasting and degradation (e.g. Iversen and Robert, 2015; Iversen et al., 2010; Laurenceau-Cornec et al., 2015; Shanks and Edmondson, 1989). In the present study, we sampled the natural phytoplankton assemblage from waters combining the highest POC concentration and chlorophyll fluorescence in an attempt to assess the aggregation potential in the most productive water strata. As $PA_r$ were formed from material at shallower depths than $PA_n$ were collected, we cannot be certain that the observed differences are not depth related. Further, we left the roller tanks to rotate for a period of 9 days in the dark and cold to simulate POC aggregation in the water column, accelerating plankton death and vulnerability to microbial degradation. These conditions deviate from natural conditions in which the plankton sinking from 12 m depth would have experienced progressive light and temperature decreases as well as changes in grazing pressure. Therefore, it is difficult to ascertain the cause of differences between roller tank formed and in situ collected aggregates, and rather we use the roller tank formed aggregates to get an estimate of the carbon-specific respiration rate of the aggregates at shallow depths.

## 4.3 Role of particle associated microbes in mesopelagic POC flux attenuation

Despite the uncertainties in the mechanisms governing rates of particle associated microbial respiration, we are still able to assess the importance of particle associated microbial respiration on the attenuation of fast sinking POC in the mesopelagic and compare our results to the small number of other recent studies (Collins et al., 2015; McDonnell et al., 2015). We calculate the flux of fast sinking POC ($F_z$) at each depth ($z$) that would result if the only loss was via particle associated microbial respiration. Calculations were based on the relationship between the remineralization length scale ($L$ in m$^{-1}$) (see Iversen and Ploug, 2013; Iversen et al., 2010), carbon-specific respiration rate ($C_{spec}$ in d$^{-1}$) and sinking velocity ($w$ in m d$^{-1}$).

$$L = \frac{C_{spec}}{W} = (-ln(F_z/F_0)/(z - z_0)) \tag{5}$$

We calculate upper and lower bounds on the remineralization length scale based on uncertainties in our measurements of $C_{spec}$ and $w$, as described in section 2.6. We compare observed fast sinking POC flux attenuation and predicted losses via particle associated microbes over two discrete depth horizons; a region of rapid attenuation of 36-128 m and slow attenuation zone of 128-500 m. Note that we exclude the non steady state value of POC flux at 500 m and instead use the value predicted from our power-law fit (Fig. 4) with bounds based on the standard deviation of our $b$ value estimated via bootstrap analysis (see section 2.7). Our data suggest that particle associated microbial respiration plays only a minor role in POC attenuation in the upper water column (8%; range: 1-14%), but becomes more important below this (33%; range: 12-50%) as the rate of POC attenuation decreases (Fig. 9a). Our measurements are based on a sub sample of the total assemblage of particles found in the water column, in particular only PA. If rates of microbial respiration are vastly different on other particle types then this would affect our calculations of POC removal by particle associated microbes. However, considering

the dominance of PA in our samples (Fig. 2), we believe our calculations reflect the bulk of the sinking material at the time of sampling. We were not able to measure respiration rates on FP due to their low numbers and small size which adds uncertainty to our estimate of the contribution of particle associated microbial respiration to POC loss. However, even if FP were respired completely, they account for less than 10% of the flux between 36-128 m and thus could not resolve the large

imbalances between POC supply and respiration that we observe in the upper mesopelagic.

Low rates of respiration result in only a very small loss of POC with depth below the euphotic zone. Thus our data agree with a recent study (Collins et al., 2015), suggesting that only a small fraction of sinking POC is removed by particle associated bacteria. Despite being hotspots for microbial activity compared to the water column (Thiele et al., 2015), particle associated microbial respiration may still be a minor contributor to the reduction in POC flux when compared to rapid loss

via processes such as zooplankton grazing and fragmentation (Dilling and Alldredge, 2000; Stemmann et al., 2000; Svensen et al., 2014). This hypothesis is supported by measurements made in the mesopelagic of the Scotia Sea on faecal pellets (Belcher et al., 2016) and on PA off Cape Blanc, Africa (Iversen et al., 2010), as well as model studies (Gehlen et al., 2006; Stemmann et al., 2004).

## 4.4 Mesopelagic carbon budget

Recent assessments of the mesopelagic carbon budget at the PAP site, although balanced over 50-1000 m, revealed an imbalance when the upper and lower mesopelagic were examined separately (Giering et al., 2014). Particle associated respiration was not directly measured in the aforementioned study, and hence we assess whether this term could help to explain observed imbalances. In this way we test whether the low respiration rates (0.001-0.173 $d^{-1}$) measured by Collins et al. (2015) are also applicable to our study site or whether the higher rates, such as observed in the western subtropical North

Atlantic gyre (0.4 $d^{-1}$) by McDonnell et al. (2015) are more appropriate. As our zooplankton net tows are integrated to 200 m, we compare sources and sinks of POC over the depth range of 36-200 m.

Although we measure low rates of both absolute and carbon-specific PA microbial respiration (<3% $d^{-1}$) suggesting that this term cannot resolve imbalances in the upper mesopelagic carbon budget, we may underestimate the importance of particle associated microbes if solubilization of POC to DOC by ecto-enzymatic hydrolysis is significant (Alldredge, 2000;

Grossart and Simon, 1998; Smith et al., 1992). This solubilization to DOC is likely to fuel the respiration of free-living microbes. Smith et al. (1992) estimated that 97% of the hydrolysates produced by bacteria in marine snow were released, with the remaining 3% being utilized by bacteria in the aggregate. However, this value was based on nitrogen-rich amino acids and hydrolysis for carbon is likely lower as it is lost more slowly than nitrogen from sinking particles. To calculate potential hydrolysis of carbon from particles, we conservatively assume a value of 75% (i.e. assuming our measured loss via

respiration is 25% of the total POC loss via particle associated microbes), which sits between Smith et al.'s (1992) value and carbon solubilization losses of <30% measured in copepod faecal pellets which are much less porous (Møller et al., 2003). With additional loss of fast sinking POC via solubilization we find particle associated microbes can explain 49% (39-57%) of POC losses in the upper mesopelagic (36-200 m) (Fig. 9b). In spite of the limitations in our estimations of solubilization it

is clear that a large discrepancy still remains in terms of an excess POC supply of 118 mg C m$^2$ d$^{-1}$ (60-177 mg C m$^{-2}$ d$^{-1}$ considering our calculated error bounds on rates of particle associated microbial respiration and fast sinking POC flux measurements) over the upper 36-200 m.

The other direct loss of POC in the mesopelagic is via zooplankton respiration and 'sloppy feeding' (cell breakage during feeding and subsequent release of DOC) (Jumars et al., 1989). Our measured zooplankton respiration rates are likely an overestimate of their contribution to POC losses in the upper mesopelagic (36-200 m) as they include both migratory and non-migratory individuals, as well as individuals above the mixed layer depth. Even with these overestimations, zooplankton respiration (5.1-10.1 mg C m$^{-2}$ d$^{-1}$) accounts for only a small POC sink in the upper mesopelagic (Fig. 9b). We are not able to account for losses of POC to DOC or suspended POC via sloppy feeding.

The direct hydrolysis by attached microbes likely supplies free-living communities with DOC (Cho and Azam, 1988; Karl et al., 1988; Kiørboe and Jackson, 2001). However by definition free-living microbes are not associated with particles and hence do not contribute directly to the loss of large fast sinking POC measured here, and as such we do not consider this loss process. The definition of dissolved and particulate is operational based on the pore size of a GF/F filter, and therefore microbes defined as 'free-living' may in fact be able to utilize colloids (Arístegui et al., 2009). However as we only measure the loss of large, fast sinking POC, we exclude free-living bacteria from our analysis of fast sinking POC loss processes. Free-living prokaryotic respiration may account for the ultimate loss of organic carbon from the organic carbon pool but we believe this is reliant on mechanical breakdown of large, fast sinking POC by zooplankton and protozoa (Lampitt et al., 1990; Poulsen and Iversen, 2008; Poulsen et al., 2011) and enzymatic hydrolysis (Smith et al., 1992). Previous measurements at the PAP site suggest that prokaryotic respiration results in loss rates of 42 mg C m$^2$ d$^{-1}$ between 36-203 m which greatly exceed estimated DOC input to the upper 1000 m (15 mg C m$^2$ d$^{-1}$) (Giering et al., 2014), supporting this hypothesis.

POC loss via zooplankton respiration, particle associated microbial respiration and solubilization, as typically invoked in model studies (e.g. Anderson and Tang, 2010) can therefore not account for observed losses of fast sinking POC in the upper mesopelagic, suggesting our knowledge of the mesopelagic carbon budget is still poorly constrained and/or incomplete.

## 4.5 The missing piece of the mesopelagic carbon budget?

Before we begin to examine whether we are indeed missing a piece of the upper mesopelagic carbon budget puzzle we must acknowledge the limitations of our estimates thus far which may in themselves rectify imbalances. Large uncertainties surround our estimates of solubilization by particle associated microbes. We would require solubilization of 87% to balance our budget which we believe to be high considering estimates of 97% for nitrogen (Smith et al., 1992) which is preferentially remineralized over carbon. In addition, as particle associated microbial respiration is able to account for a greater proportion of the fast sinking POC loss in the mid mesopelagic (200-500 m), solubilization losses would need to be lower over this depth region (66%) to maintain a balance. We are not aware of any studies suggesting that rates of

solubilization would vary with depth. Increased solubilization would present itself in the form of increased DOC and/or increased rates of microbial respiration, however these terms are included in the estimate by Giering et al. (2014) and a large imbalance in the upper mesopelagic is still apparent in their budget. We are not able to rule out increased solubilization in the upper mesopelagic as an additional sink term, but consider it unlikely to solve the imbalance.

5       Although our method of measuring particle associated microbial respiration attempts to avoid bottle effects and accurately simulate the environment of a sinking particle, we were not able to simulate pressure changes. Colonization of particles by pressure adapted microbial communities at depth may lead to an underestimation of in situ microbial activity using decompressed samples (Tamburini et al., 2013). Recent work suggests that the attached microbial community on sinking particles is 'inherited' from the fluorescence maximum (Thiele et al., 2015); these organisms are not adapted to

changes in pressure (Tamburini et al., 2006, 2009) and temperature, and therefore exhibit lower prokaryotic growth efficiencies (PGE) and overall metabolic rates. Similarly, our experiments were carried out at constant temperatures whereas particles sinking through the water column will experience the range of water column temperatures, likely impacting all metabolic processes. We are also limited in our study by the lack of replicate MSC deployments at each depth. Although numerous aggregate respiration rates were measured from each sample, high patchiness in the type and source location of

sinking particles could result in greater variability in respiration rates.

      Additionally we are not able to measure mechanical disaggregation via processes such as fluid shear which could provide additional losses of large sinking POC. The forces required to break apart large marine snow aggregates have been shown to be higher than typical estimates of energy dissipation in the ocean, suggesting that this would not be a major loss process (Alldredge et al., 1990). Physical disaggregation could be more important in surface waters where dissipation rates

can exceed the forces required to break marine snow aggregates (Alldredge et al., 1990; Burd and Jackson, 2009). However, we suspect that only a small fraction of sinking POC would be fragmented by abiotic processes to particles <0.15mm and hence would not explain the large loss of fast sinking POC measured in this study.

      In order to address imbalances in the sources and sinks of fast sinking POC to the upper mesopelagic we require an additional loss process of POC. One key term missing from the budget is that of free-living protozoans which would not be

collected in zooplankton nets and can make up a substantial part of marine planktonic ecosystems (Biard et al., 2016). Laboratory experiments on copepod FP reveals that dinoflagellates degraded FP over three times faster than bacteria (0.18 d$^{-1}$ compared to 0.04 d$^{-1}$), and the combined effects of bacteria, dinoflagellates and copepods led to FP degradation rates of 1.12 d$^{-1}$ (Svensen et al., 2014). Dinoflagellates and ciliates have been shown to feed on fecal pellets (Poulsen and Iversen, 2008; Poulsen et al., 2011) and PA (Tiselius and Kiørboe, 1998). Therefore POC loss via protozoan respiration may account

for at least some of the additional POC loss we require to resolve imbalances in upper mesopelagic carbon budgets.

      Loss of sinking POC via fragmentation of large sinking particles into small (<0.15 mm ESD) and non-sinking particles by both abiotic and biotic means may also explain some of our observed imbalance in the upper mesopelagic as the POC fluxes measured here are for 'fast sinking' particles only (see methods). We hypothesize, in line with a growing number of other studies (Cavan et al., 2015; Collins et al., 2015), that zooplankton living in the upper mesopelagic may

stimulate this loss in POC via fragmentation from sloppy feeding, swimming activities and/or microbial gardening (Iversen and Poulsen, 2007; Mayor et al., 2014). Fast sinking particles can reach the deep ocean with minimal degradation due to slow rates of particle associated respiration. Conversely, once fragmented, the increased residence times (in terms of their sinking rate) of slow and non-sinking POC could allow a sustained loss of POC over the season by microbial respiration. In theory, this seasonal balance should present itself in the form of an increase in slow and non-sinking POC following the seasonal peak in fast sinking POC and a more gradual decline over the season. This less rapid seasonal decline in slowly sinking material is apparent in the results of a biogeochemical model study for subpolar regions (Henson et al., 2015). Seasonal cycles in POC (0.8-200 µm via GF/F filtering) have been detected following analysis of long term time series data at station ALOHA in the Pacific (Hebel and Karl, 2001). They suggest that the build up and removal of standing stocks of POC do not require a large degree of decoupling between production and loss processes and can exist due to small but sustained differences. Considering the highly dynamic nature of the typical bloom-bust scenario of the North Atlantic it seems unlikely that a balance in source and sink processes would be found by 'snapshot' measurements such as made here.

Additional inputs/losses of organic carbon could be driven via physical processes such as advection or changes in mixed layer depth (e.g. Dall'Olmo and Mork, 2014). Although the mixed layer was relatively stable during our study period, the winter deepening in MLD to 250 m (Hartman et al., 2015) could provide a seasonal balance to the budget if concentrations of slow and non-sinking particles are sufficiently high (Bochdansky et al., 2016). Similarly, advective processes are an unaccounted for source/sink of carbon in this study and could result in closer agreement between sources and sinks.

## 5 Conclusions

We present here a unique vertical profile of particle associated microbial respiration measured directly on sinking marine aggregates collected at depth. Rates of carbon-specific respiration were relatively constant with depth, and particle associated microbial respiration amounts to a small loss term in the mesopelagic carbon balance. We suggest that it may be possible to explain the loss of fast sinking particles (>0.15 mm ESD) in the upper mesopelagic through a combination of particle associated microbial respiration, solubilization, and the conversion into small (<0.15 mm ESD) and non-sinking POC via zooplankton and protozoan mediated processes. Material lost through fragmentation would be retained in the upper mesopelagic allowing it to be slowly respired over time and enabling a balance of the mesopelagic carbon budget only over seasonal timescales. However, detailed information about fragmentation processes are lacking and are needed to better constrain the upper mesopelagic carbon flows. Moreover, there is a need for seasonally resolved studies (of fast and slow sinking pools of carbon) to get a better appreciation of how changing primary production in a non-steady state system can influence seasonal fluxes of POC in the mesopelagic.

**Acknowledgements**

We would like to thank the officers, crew and science party aboard RRS Discovery for cruise DY032. In particular we would like to thank Emma Cavan and Chelsey Baker for their invaluable help with snow catcher deployments. In addition we would like to thank Christian Tamburini for his advice at early stages of manuscript preparation. This work was supported through Natural Environment Research Council National Capability funding to Stephanie Henson and Richard Sanders, National Oceanography Centre, as well as a studentship to Anna Belcher from the University of Southampton. This work is also a contribution to the Mediterranean Institute of Oceanography "AT-EMBE", and to the Labex OT-Med (ANR-11-LABEX-0061, www.otmed.fr) funded by the Investissements d'Avenir, French Government project of the French National Research Agency (ANR, www.agence-nationale-recherche.fr) through the A*Midex project (ANR-11-IDEX-0001-02), funding Virginie Riou during preparation of the manuscript.

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

**Figures and Figure Legends**

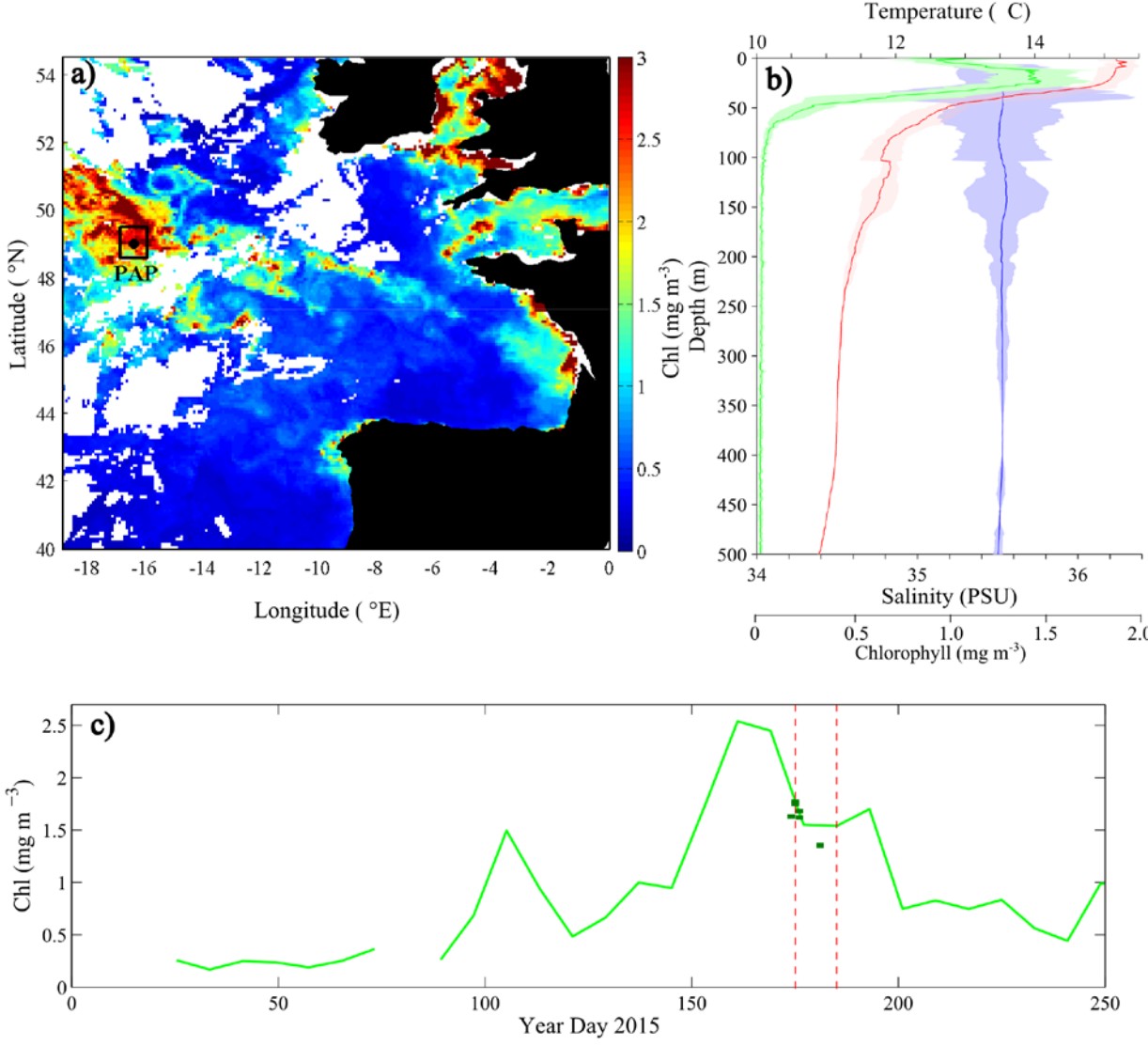

Figure 1: Surface chlorophyll concentration (mg m[-3]) at the PAP site, (a) PAP study region (black box) overlain on 9 km Aqua MODIS satellite chlorophyll for 18/06/2015-25/06/2015. (b) Average vertical temperature, salinity and chlorophyll profiles measured at the PAP site (red, blue and green lines respectively) and the standard deviation (light shading) of CTD deployments coinciding with MSC deployments. (c) Temporal change in surface chlorophyll (mg m[-3]) over the PAP study region based on 8 day, 9 km Aqua MODIS satellite data. Gaps in data are due to cloud cover. Vertical red lines indicate start and end of sampling period, and dark green squares are discrete measurements made from the CTD at depths of 5-10 m.

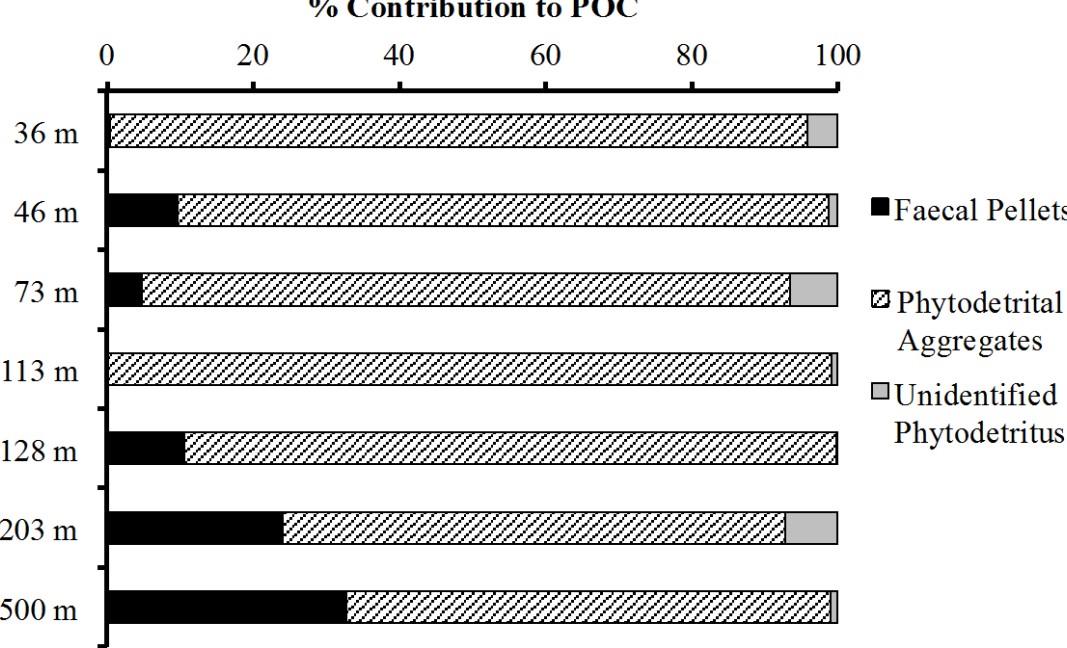

**Figure 2: Composition of fast sinking POC at each measured depth horizon. The percent (%) contribution of faecal pellets (black), phytodetrital aggregates (hatched), and unidentified phytodetritus (grey) to the total mass of fast sinking POC collected in Marine Snow Catchers at each depth horizon. See Table 1 for numbers of particles in each category.**

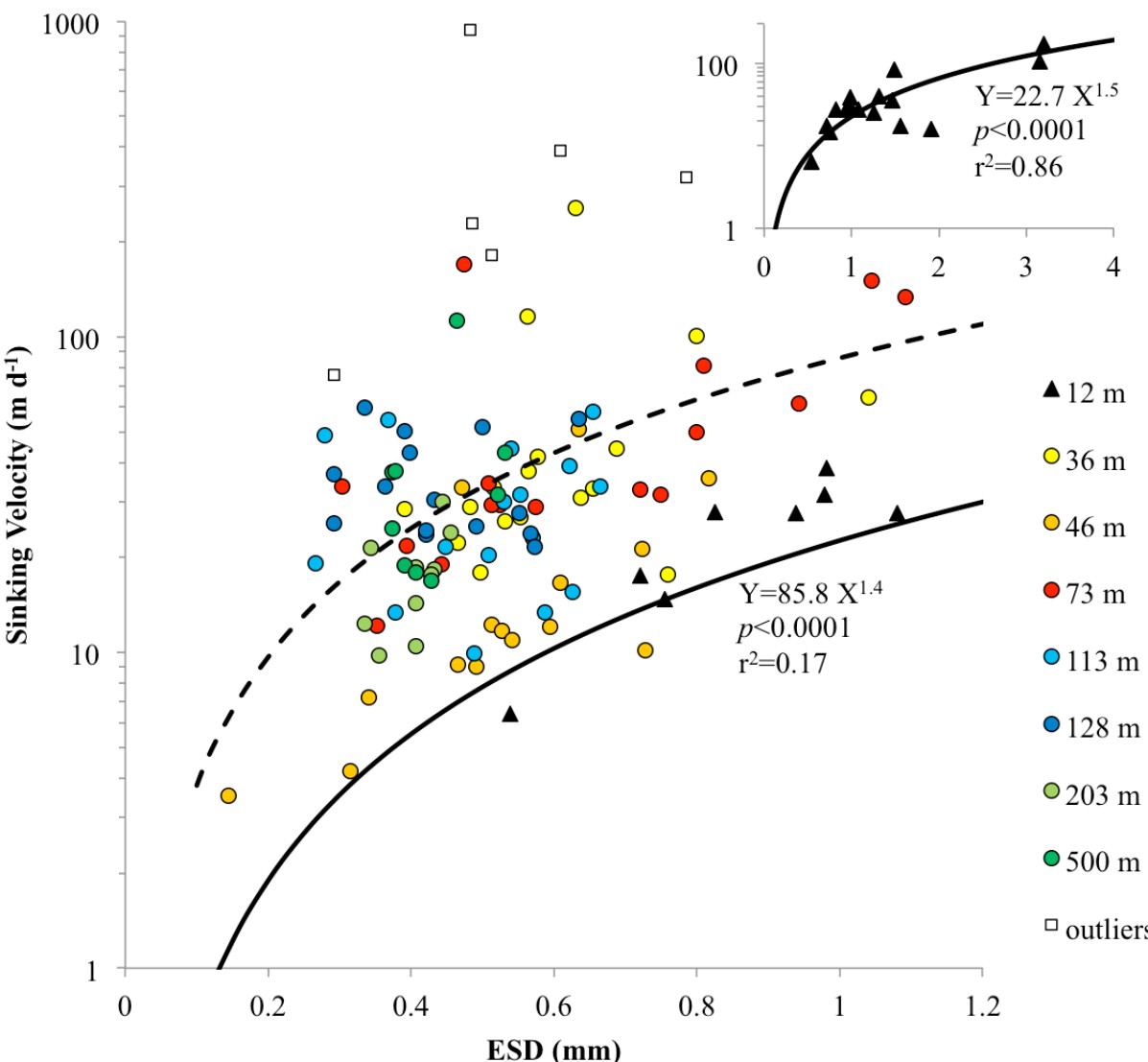

**Figure 3: Relationship between sinking velocity (m d⁻¹) and equivalent spherical diameter (ESD, mm) of phytodetrital aggregates.** Roller tank formed aggregates are shown by triangles (black) and natural aggregates collected at depth by circles coloured by depth (36 m=yellow, 46 m=orange, 73 m=red, 113 m, light blue, 128 m=dark blue, 203 m=light green, 500 m=dark green). Note the log scale on the Y axis. We apply a power-law fit for roller tank (solid line $Y = 22.7X^{1.5}$) and natural (dotted line $Y = 85.8X^{1.4}$) aggregates. Inset displays roller tank particle data only showing full size range. 6 outliers (black open squares), defined as being outside 2 standard deviations from the mean, were excluded in the relationship for natural aggregates.

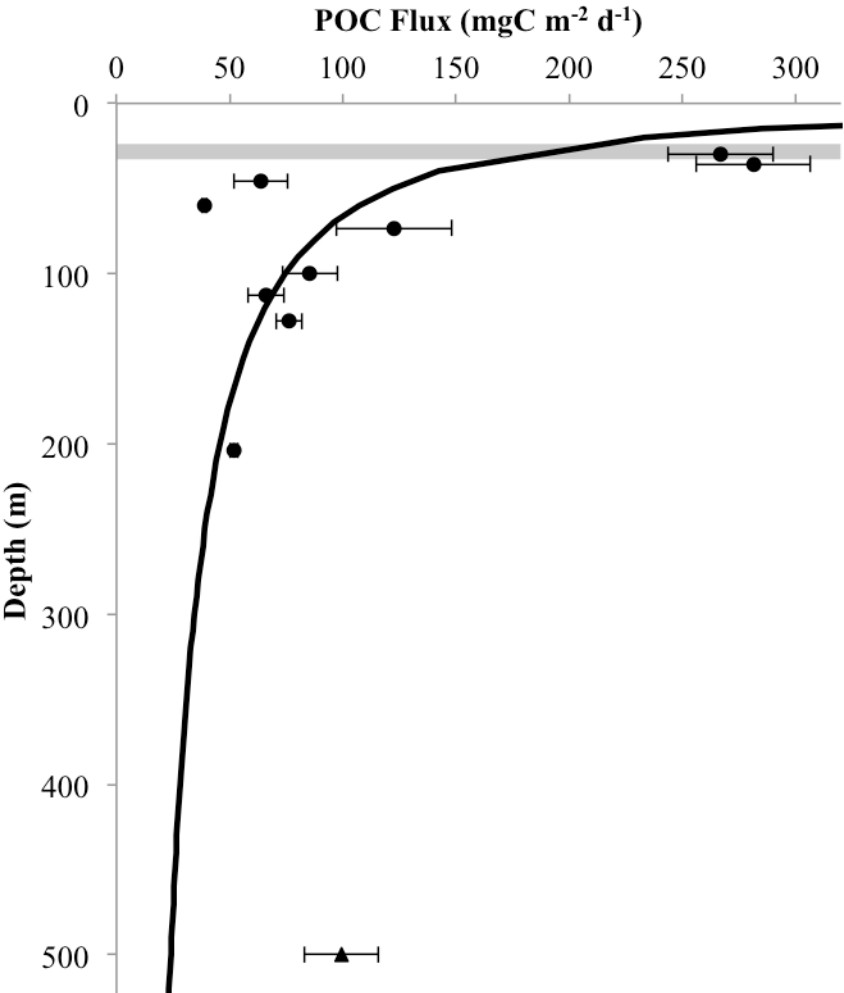

**Figure 4: Flux of POC (mg C m$^{-2}$ d$^{-1}$) with depth at the PAP site. POC fluxes of fast sinking particles measured in June 2015 at the PAP site via deployment of Marine Snow Catchers. Error bars relate to duplicate filters per sample. A power-law curve was fitted to the data (black line), Y=194.9 ·(X/MLD)$^{-0.71}$ ($R^2$= 0.42, $p$=0.060, n=9), excluding the point at 500 m (triangle) which is likely due to non steady state conditions. The grey shaded area indicates the mixed layer depth over the study period.**

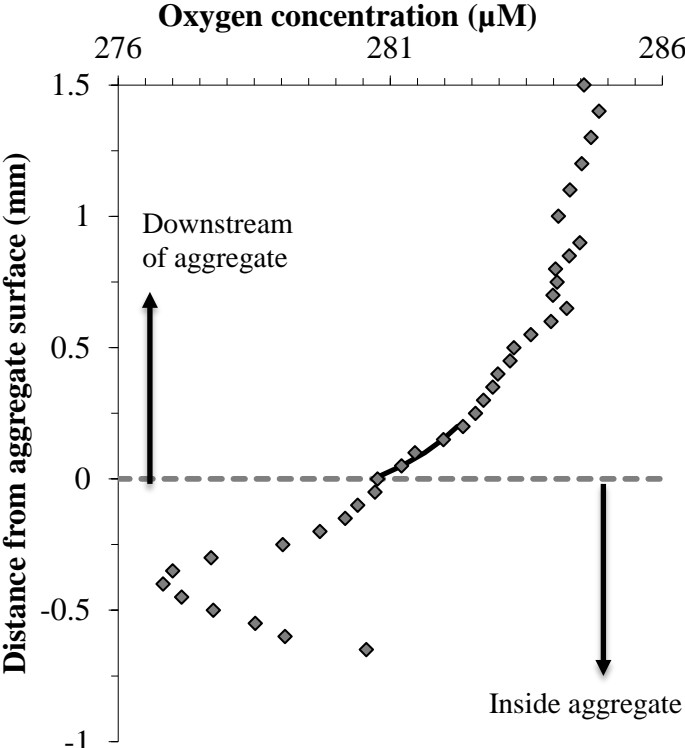

**Figure 5: Example oxygen profile (µM) through a phytodetrital aggregate collected at 46 m depth. Measurements were made with microsensors in steps of 50-100 µm, with negative values reflecting the distance into the aggregate from the surface. The solid black line shows the model fit used to calculate the oxygen flux in the diffusive boundary layer.**

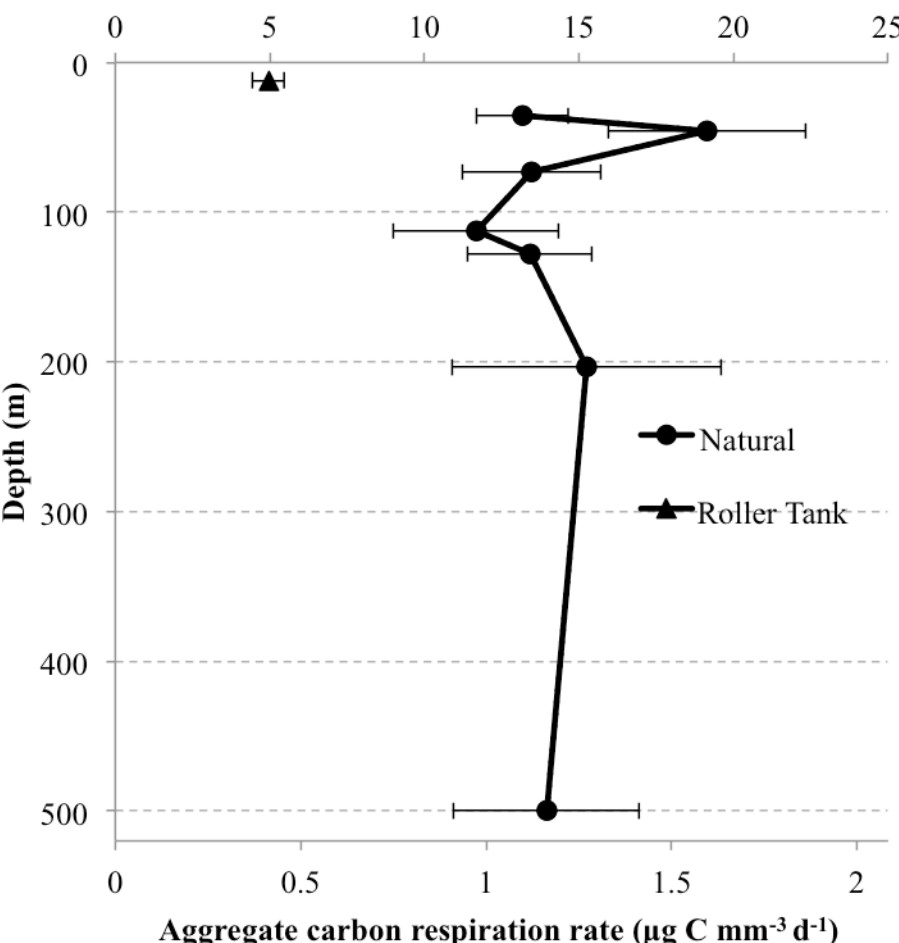

**Figure 6: Respiration rates of phytodetrital aggregates with depth. Oxygen fluxes to aggregates (nmol O$_2$ mm$^{-3}$ d$^{-1}$) for natural aggregates collected at depth (black circles, solid black line) and roller tank formed aggregates (black triangle). For reference aggregate respiration rates are also shown in terms of carbon per aggregate volume (μg C mm$^{-3}$ d$^{-1}$). Data are for experiments carried out at 10 °C. Error bars represent +/- 1 standard error.**

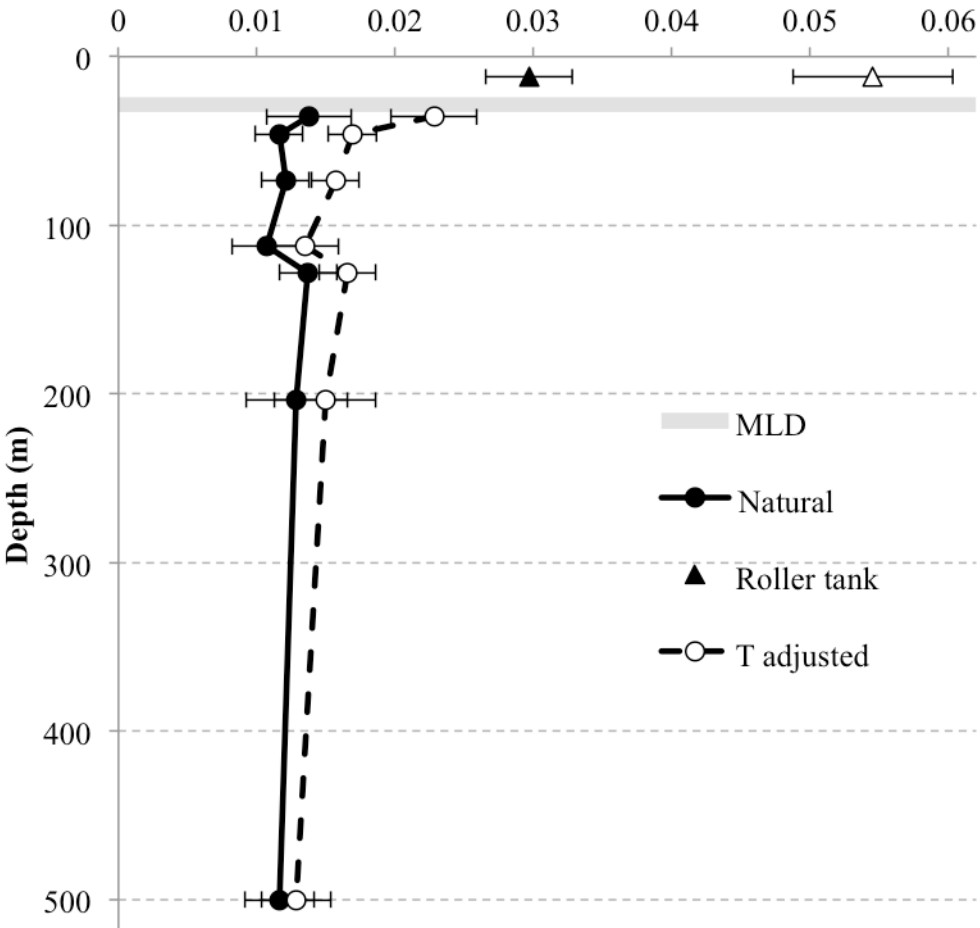

**Figure 7: Carbon-specific respiration rates (d⁻¹) for natural aggregates collected at depth (black circles, solid line) and roller tank (triangles) formed aggregates. Rates adjusted to the in situ temperature (T) are shown by the dashed black line (open circles and open triangle for roller tank). Grey shading shows the range in mixed layer depth over the study period, and error bars represent standard errors.**

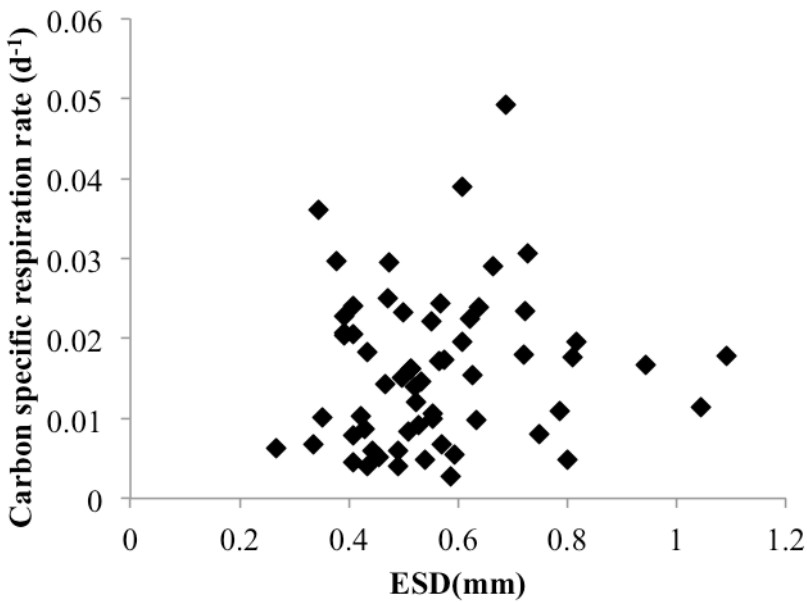

**Figure 8: Carbon-specific respiration rates (d⁻¹) of natural phytodetrital aggregates collected at depth. Data have been adjusted for in situ temperatures (see text).**

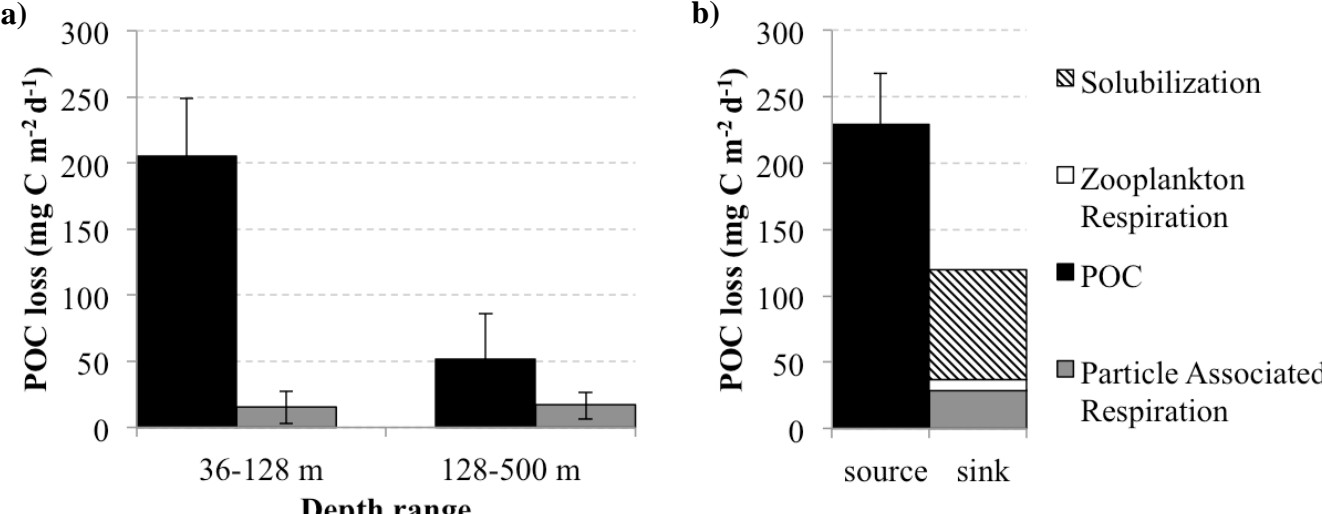

**Figure 9: Balance of processes controlling POC flux attenuation. (a) Comparison of observed POC loss (black bars) and estimated POC loss based on particle associated microbial respiration only (grey bars) over two depth horizons (36-128 m, and 128-500 m). (b) Assessment of POC sources and sinks in upper 200m. Additional estimated losses via solubilization by particle associated microbes (calculated assuming respiration accounts for 25% of the total loss by particle associated microbes and solubilization the remaining 75%, see section 4.4), and zooplankton respiration based on integrated (0-200 m) net data. Error bars in figure (a) relate to the range in POC flux measured at each depth and the range in respiration rates calculated (see section 2.6). We do not display error bars on our assessments of POC sinks due to the potentially large and unconstrained errors on solubilization.**

**Tables**

**Table 1: Deployment table for cruise DY032 to the PAP site.**

| Depth (m) | Date | Time (GMT) | POC Flux (mg C m$^{-2}$ d$^{-1}$) | # PA (in 95 L sample)** | # FP (in 95 L sample)** | Average PA ESD (mm) | Median PA sinking rate (m d$^{-1}$) |
|---|---|---|---|---|---|---|---|
| 36 | 24/06/2015 | 02:10 | 281.4 | 785 | 23 | 0.60 | 33.2 |
| 128 | 24/06/2015 | 02:35 | 76.2 | 259 | 61 | 0.53 | 11.7 |
| 73 | 26/06/2015 | 14:45 | 122.8 | 252 | 15 | 0.65 | 34.0 |
| 113 | 28/06/2015 | 11:50 | 66.0 | 198 | 0 | 0.49 | 30.1 |
| 500 | 28/06/2015 | 17:50 | 99.4 | 282 | 92 | 0.44 | 30.6 |
| 46 | 30/06/2015 | 18:45 | 63.7 | 702 | 61 | 0.41 | 18.3 |
| 204 | 02/07/2015 | 10:00 | 51.8 | 275 | 76 | 0.44 | 34.4 |
| 30 | 04/07/2015 | 13:00 | 266.8 | | | | |
| 60 | 04/07/2015 | 13:20 | 38.7 | | | | |
| 100 | 04/07/2015 | 13:30 | 85.5 | | | | |
| 12* | 24/06/2015 | 00:38 | | | | 1.39 | 29.4 |

PA: Phytodetrital aggregate; FP: Faecal Pellet; ESD: Equivalent spherical diameter

* Roller tank data: Deployment date, time and depth refer to CTD cast from which water for roller tank experiments were obtained

**Refers to counts of fast sinking material collected from deployment of 95 L snow catcher bottle. Counts have been scaled up from smaller sample split (1/4).

**Table 2: Rates of particle associated microbial respiration rates in phytodetrital aggregates. Averages are given for each depth with full range in brackets. Results are for experiments carried out at 10 °C.**

| Depth (m) | Total $O_2$ consumption (nmol $O_2$ agg$^{-1}$ d$^{-1}$) | Volumetric $O_2$ consumption (nmol $O_2$ mm$^{-3}$ d$^{-1}$) | $C_{resp}$ (ng C mm$^{-3}$ d$^{-1}$)* | $C_{spec}$ (d$^{-1}$) ** | # aggregates measured |
|---|---|---|---|---|---|
| 12 ^ | 12.62 (1.23-62.80) | 4.96 (2.48-7.65) | 0.059 (0.030-0.092) | 0.030 (0.012 -0.054) | 12 |
| 36 | 1.25 (0.56-2.81) | 13.18 (8.03-17.77) | 0.158 (0.096-0.213) | 0.014 (0.006-0.030) | 6 |
| 46 | 2.04 (0.65-3.89) | 19.12 (9.49-43.76) | 0.230 (0.114-0.525) | 0.012 (0.004-0.021) | 10 |
| 73 | 2.46 (0.22-6.92) | 13.47 (4.69-37.67) | 0.162 (0.056-0.452) | 0.012 (0.004-0.030) | 15 |
| 113 | 0.98 (0.07-2.80) | 11.66 (3.10-32.39) | 0.140 (0.037-0.390) | 0.011 (0.002-0.024) | 10 |
| 128 | 0.93 (0.20-1.88) | 13.40 (3.22-19.73) | 0.161 (0.039-0.237) | 0.014 (0.003-0.020) | 8 |
| 204 | 0.46 (0.14-0.87) | 15.25 (4.05-36.77) | 0.183 (0.049-0.441) | 0.013 (0.003-0.031) | 7 |
| 500 | 0.74 (0.17-1.24) | 13.95 (4.86-24.71) | 0.167 (0.058-0.297) | 0.012 (0.004-0.021) | 5 |

^ Roller tank

* Volume specific respiration rate ($C_{resp}$)

5 **Carbon-specific respiration rate ($C_{spec}$)