# Peer review of "Depth-resolved particle associated microbial respiration in the northeast Atlantic"

_Biogeosciences, 2016_

## Referee Comment (RC1) · Anonymous Referee #1 · 10 May 2016

Summary: The authors use a combination of emerging and established methods to address a major outstanding question in chemical oceanography; namely, the degree to which particle flux attenuation (i.e., the progressive remineralization of sinking marine particles with depth) is a result of respiration by attached bacterial communities, compared with various other biological and abiotic processes. The authors have a robust and very welcome dataset that has the potential to contribute significantly to a growing body of work in this area. Using shipboard measurements of respiration on particle material retrieved from depth, the authors conclude (in agreement with several other very recent studies) that direct respiration by particle-attached bacterial communities can explain only a fraction of the attenuation that is observed in the environment using sediment traps. (The relative importance of attached bacterial respiration appears to increase with depth compared with other processes.) The other major (and truly novel)

[Figure]

finding of the paper – which is not presently addressed in the abstract, but should be – is the very apparent and mysterious difference in amenability to heterotrophic respiration between actual sinking marine particle material and artificial aggregates created in roller tanks. This is striking, because the latter have been invoked in dozens of experiments over the past 20 years as proxies for true sinking particle material. In the present study, the labilities of the two types of particles appear to differ very significantly.

General impression and recommendation: This is scientifically interesting work that deserves publication in Biogeosciences. The conclusions advance the dialog incrementally in one avenue of research (the partitioning of particle flux attenuation between attached respiration and other processes) and present a very new and striking finding with regard to the roller tank particles versus "real" particles. The authors' central findings and conclusions are acceptable and meet the criteria for publication in Biogeosciences. However, I have some very significant concerns which pertain primarily to (1) the authors' reading, interpretation, and presentation of the existing literature and state of the art as regards the biological pump (several issues), (2) the interpretation of respiration rates obtained using the authors' method, and (3) the structure of their manuscript. The authors could address these concerns with some re-analysis (particularly a more diligent consideration and assimilation of uncertainties), a more careful (and nuanced) interpretation of the data, to include mention in the manuscript of some very relevant caveats, and some re-writing of the manuscript, in places. In addition, important details of calculations and mathematical methods are omitted.

The manuscript is generally well-written and the figures are generally clear. I have some suggestions for the authors on two of their figures; addressing these should be trivial.

General comments:

(1) In both the abstract and body of the manuscript, the authors generally overstate the novelty of their finding with regard to the importance of particle-attached respiration.

As they note on pp. 11-12, a small number of other recent studies have also presented similar results. I would urge the authors' to revisit the manuscript and ensure its tone reflects the findings in these existing studies apart from the citations on pp. 11-12.

(2) I have some significant concerns about the degree of handling, size-fractionation, and manipulation involved in the collection and incubation of the particle material in this study. First, the authors invoke the term in situ for these measurements; this is neither correct nor appropriate. Second, I am concerned that the particle material used for the incubation studies was manipulated and size-fractionated to a degree that makes comparison of respiration rates measured on the material to the MSC sediment trap flux measurements very questionable (specific concerns below).

(3) The authors' findings regarding the difference in lability between the two types of particles is really the novel contribution of this paper; I would encourage them to revisit the manuscript with an eye toward elevating the importance of these conclusions.

(4) The authors do not describe at all their methods for error assimilation and propagation; in fact, it is unclear whether this was even considered. As the authors are aware, the business of extrapolating rates of respiration on single particles in a highly controlled environment to the carbon cycle of a marine water column involves a large number of calculations involving a number of conversion and scaling factors, each of which have their own uncertainties. I would like to see some sort of more robust error propagation/analysis in the manuscript.

Specific comments:

Abstract generally: Why no mention of the roller tank versus "real" particle respiration rate comparison?

p. 1, lines 11-12: Logic suggests the first sentence of the abstract should be restated in the converse, i.e., "Atmospheric levels of carbon dioxide are tightly linked to the depth at which sinking POC is remineralized in the ocean."

[Figure]

p. 1, lines 17-18: I am confused by the authors' statement of their hypothesis here. Perhaps they mean something more specific, i.e., "... the missing sink for particulate carbon in the upper mesopelagic"? In the growing number of recent studies in which other oceanographers have been unable to close mesopelagic carbon budgets (those in which both particulate and dissolved/suspended water column phases have been considered together), the problem has been one of an apparent undersupply of total carbon to the system, not a problem of too little respiration. However, if the authors are only considering the particulate phase, then this supposition would make more sense since the studies they reference on pp. 11-12 have demonstrated the opposite is true when considering the sinking particulate phase exclusively.

p. 1, line 15: Confusing as currently stated. Try: "Attempts to balance POC supply to discrete depth layers of the mesopelagic..." (if that is in fact what the authors mean)

p. 1, line 19 ff: Use of in situ confusing and inappropriate. If the authors mean here that the particles were collected from depth, then that should be stated; anything collected from the ocean in the course of oceanographic research can be said to have been collected "in situ." The respiration measurements the authors make this study are not in situ measurements; the only true in situ particle respiration measurements that I know of are those of McDonnell et al. (2015), obtained using the RESPIRE device. The measurements in the present study, as in Collins et al. (2015), are shipboard measurements made using material collected from depth.

p. 1, line 25: This shift could also be ultimately to DOC (not simply to suspended POC).

p. 1, line 30: This is an incorrect interpretation of Martin's "b"; b does not represent the depth at which material is remineralized. (In the length-scale-based parameterization the authors invoke on p. 11, the interval z-z0 represents the depth interval over which material is remineralized.) This is one of the limitations of the Martin formulation, since the exponent doesn't really have any explicit meaning.

p. 3, lines 3-4: I am not sure this is true; McDonnell et al. and Collins et al. both

made profiles of particle-associated respiration rates using actual particle material. Is it the authors' contention that their use of "individual" particles makes this the first such study? Seems like a rather qualified claim to novelty that could be omitted from the paper without affecting its impact or conclusions.

p. 3, line 15: How did this measure of MLD compare to the 1% PAR level (traditionally invoked as the base of the euphotic zone, which is the reference depth for most other particle flux studies).

p. 4: I have some concerns about the representativeness of the respiration rates obtained from the authors' method, which appeared to require very extensive handling and manipulation in the selection and isolation of individual particles. Wouldn't this method of "plucking" by eye perhaps result in separation of the particles from the microbial communities with which they were likely associated in the environment? Further, how do the authors justify the assumption that the sum of the individual rates obtained on a few chosen particles can be applied to estimate rates of removal vis-a-vis the complete, heterogeneous collection of particles in the corresponding MSC fractions used for flux determination? Would it not have been better to measure rates on an assemblage of particles that was (a) subjected to less handling and (b) was more completely representative of the wider spectrum of shapes and sizes of the particles in the traps? I am not suggesting the errors introduced by this approach render it invalid (particularly given the enormous uncertainties involved in every other aspect of this sort of work), but a more complete discussion of the implications of their decision would be welcome.

p. 5, line 25: Conversions of PA volume to what?

p. 5, line 30: The way the authors have currently defined Cresp (line 18) and [POC] (line 29), it seems to me this equation yields a quantity Cspec with units of length (m) per unit time, not simply time-1. Perhaps the authors can clarify?

p. 6, line 6: What is meant by "exponential fit" (i.e., what is the form of the equation)?

p. 6, line 25: Are the 1.9 and 1.4 mg m-3 figures from discrete measurements or ocean color data?

p. 7, lines 7-10: Depending on the time of day at which the traps were collected, this increase in FP numbers could also be aliasing daily vertical migration.

p. 9, lines 16-17: I do not understand the authors' meaning here. In addition, what is meant by "similarity"; this is a nonspecific and vague term.

p. 9, lines 19-20: Version of record of this paper is a 2015 date of publication.

p. 9, last paragraph: Given the very significant influence of temperature on metabolism, why were the particles incubated at a static 10?C rather than at in situ temperature? If because only a limited number of incubators were available for large number of particles, this should be stated on p. 4.

p. 11, first full paragraph: This is the most intriguing and novel finding of the authors' study, and should be given more prominence.

p. 11, line 29: "Slow" compared to what? I would suggest the rates of respiration measured by others in water column samples are truly "slow" compared with particle-attached/associated rates.

p. 12, line 19: What are the error bounds on this 139.3 mg C m-2 d-1 figure?

p. 13, first paragraph: The authors do not mention the additional possible mechanical processes of shear or turbulent disaggregation.

p. 13, line 33: I concur wholeheartedly! The authors would do well to make a connection here to very recent and compelling work by Biard et al. (2016), showing the ocean may contain large numbers of these protists.

pp. 14-15: I commend the authors for their thoughtful and excellent discussion here.

p. 14, line 30: Use of in situ misleading and redundant.

[Figure]

Comments on figures and tables:

Figure 1: Panel (c) might be improved by the superimposition of some chlorophyll concentrations measured concurrently in the water samples.

Figure 2: From inspection of the figure, it seems to me the authors' decision to use only PA-derived rates (and not also FP-derived rates) is valid only for depths < 113 m. Below this depth, FP constitute a very significant fraction of the POC.

Figure 3: An excellent figure. Very clearly and compelling presents the most interesting conclusion of the authors' manuscript. However, could the authors provide the R2 for these fits in addition to the p-values?

Figure 4 (and corresponding presentation of this curve fit in the manuscript): What are the error bounds on the fitted parameters?

Figures 6, 7: Use of "in situ" to characterize the rates presented in these figures is particularly misleading.

Figure 9: In line with my concern regarding the compatibility of the authors' respiration rates with their trap flux data: Is the "aggregate POC" represented by the black bars the same POC as that supplied in the incubations used to determine the respiration rates? Seems to me this is not the case. I am not sure these can be directly compared in such a way. I am not sure I understand the meaning of the error bars ("upper" and "lower" estimates; are these error bounds obtained from some sort of uncertainty analysis?). Specifically, how were the error bars on the solubilization rates obtained, since these were based on a very hypothetical assumption?

References cited in review:

Biard, T.; Stemmann, L.; Picheral, M.; Mayot, N.; Vandromme, P.; Hauss, H.; Gorsky, G.; Guidi, L.; Kiko, R.; Not, F. 2016. In situ imaging reveals the biomass of giant protists in the global ocean. Nature 532, 504-507.

Collins, J. R.; Edwards, B. R.; Thamatrakoln, K.; Ossolinski, J. E.; DiTullio, G. R.; Bidle, K. D.; Doney, S. C.; Van Mooy, B. A. S. 2015. The multiple fates of sinking particles in the North Atlantic Ocean. Global Biogeochem. Cycles 29, 1471-1494

McDonnell, A. M. P.; Boyd, P. W.; Buesseler, K. O. 2015. Effects of sinking velocities and microbial respiration rates on the attenuation of particulate carbon fluxes through the mesopelagic zone. Global Biogeochem. Cycles 29, 2014GB004935

---

## Referee Comment (RC2) · Anonymous Referee #2 · 16 May 2016

SUMMARY

This manuscript tackles a major and current question in the understanding of biogeo-chemical cycles in the ocean regarding the processes which affect the remineralisation length scale in the twilight zone and influencing the efficiency of carbon sequestration in the deep ocean. More specifically, the authors address the question of the relative influence of particle-attached bacterial remineralisation to other processes responsible for Particulate Organic Carbon (POC) flux attenuation in the upper mesopelagic zone. A large range a methodologies are used and combined to test their main hypothesis that the loss of organic carbon due to bacterial communities attached to 'fast-sinking parti-cles' is the missing term that may help to close the carbon budget in the mesopelagic. Most of the methods used are recent but not novel and proved to be robust in previ-ous published studies. The choice of the study site located in the Porcupine Abyssal

[Figure]

Plain (PAP) in the North Atlantic is motivated by an extensive and recent pre-existing literature in this area. The resulting dataset is certainly valuable and could contribute to the scientific community. Results of their measurements combined with numerous calculations – and assumptions where needed – suggest that particle-associated bacterial respiration is not sufficient to explain the missing loss of organic carbon in the upper mesopelagic. As an alternative hypothesis, the authors propose that organic carbon losses due to the fragmentation of large fast-sinking particles into smaller slow-sinking/suspended particles operated by zooplankton in the mesopelagic could be the main process to account for to explain the imbalances observed in the mesopelagic carbon budget. Apart from a few sections, the manuscript is well-written and easy to read. The figures are clear and well presented, but in some places modifications are needed to improve the messages intended.

GENERAL COMMENTS AND RECOMMENDATION

Overall, this manuscript leaves a good impression on the goals targeted and the amount of work achieved. However, there is a striking mismatch between the hypothesis tested, the type of measurements conducted and the conclusions developed. I acknowledge the large number of measurements and understand the challenge presented by onboard analyses and that compromises have always to be made in the choice of the parameters measured but the conclusions drawn in a study have to align with the data acquired and it is unfortunately not the case here. In order to properly test the relative importance of the respiration associated with particle-attached bacterial communities, a complete set of the other known processes responsible for organic carbon loss, along with a comprehensive inventory of carbon sources should have been measured. This work lacks measurements of major parameters essential to establish a valid carbon budget in the mesopelagic. As stated in the title of the last section of the Discussion ("The missing piece of the mesopelagic budget"), the authors make an attempt to find a missing piece of a puzzle which seems to already miss several other very important pieces. In particular,

(1) the values of zooplankton abundance and respiration rates used in the calculations are inferred from another study carried out in the same area but 6 years before. To justify the validity of applying these external data to the present work, the authors state that sampling was conducted in both studies at the same stage of the seasonal cycle but no evidence are given to support it. For instance a measure of phytoplankton cell physiological state (Fv/Fm), phytoplankton community structure (often displaying the same successions each year), nutrient levels potentially indicating exhaustion, and zooplankton community structure could have provided some element of response.

(2) no estimation of the respiration due to free-living bacterias in the water column is made. Previous studies noted however that it may be the predominant term in total bacterial respiration (see Extended Data Figure 6, Giering et al., 2014).

(3) the measurements of bacteria abundance in aggregates is also missing although it would have bring essential information to conclude on the observed variations of POC content and calculated specific respiration rates in the marine snow aggregates.

(4) Only the 'fast-sinking' fraction of the particle flux is analysed. This is a major flaw in the study. While the 'slow-sinking' fraction might poorly contribute to the deep carbon export as noted by Riley et al. [2012], it is precisely because most of this fraction of the flux appears to be remineralised in the mesopelagic zone. It is very surprising to me that no total POC measurements from the MSC collection have been done to allow a comparison with POC contents measured in the aggregates selected for the analysis.

(5) no measurements of DOC was made even if it represents a non-negligible source of carbon to depth and surely needs to be accounted for in any carbon budget. Having this term would have informed – and possibly confirmed– some of the numerous hypotheses postulated in the Discussion section.

In a very honest approach, the authors detail all these missing terms in the last section of the Discussion and try to weigh their potential influence based on the literature. Unfortunately, no valid conclusion can be drawn based on so many assumptions and by

using values which are themselves subjected to very large uncertainties and potentially not applicable in this system. As a result, the main conclusion – in fact an alternative hypothesis – is disappointing since it echoes the conclusion made by Giering et al. [2014], but without bringing any additional piece of evidence to confirm it. Despite all these caveats, I believe that this work is worthy of being published, mostly for its very valuable dataset. However, a complete restructuring of the manuscript is required. The formulation of new hypotheses more in-line with the measurements conducted should help in this difficult task. The objectives initially aimed in this work (i.e. to close the carbon budget in the mesopelagic zone) have to be downgraded, but it should not minimise the importance of the findings centred on the variations of particle-associated bacterial respiration as a function of depth. As already noted in the title, I suggest to present this work as a focus on this single parameter of importance: the depth-resolved particle-associated microbial respiration in the northeast Atlantic. I recommend to highlight the variations of respiration observed at each depth and in each set of aggregates from the same depth. A large fraction of the discussion is already framed around this aspect, but a thorough evaluation of the variability attributable to errors inherent to each measurement should be examined before trying to explain the potential real variability. It might appear that propagation of the uncertainties could alone explain the variability observed at a given depth, allowing then to fully explore the depth-related variability. Finally, I wonder if the roller tank aggregation experiment is a valuable addition to this work or at the opposite if it weakens the study by showing results that contradict those made in the water column. Roller tank-made aggregates have been excluded a long time ago as models to quantify processes in real particles [Jackson, 1994]. In addition, the choice of settings used in the roller tank experiment and the methods conducted are highly questionable (see specific comments), and any evidence based on these results should be taken very carefully.

SPECIFIC COMMENTS

Abstract, line 16: "...with respiration sometimes being 50% lower than apparent carbon

loss in the upper mesopelagic." This sentence is confusing and possibly incorrect. Did the authors mean 50% "higher" rather than "lower"? Most of the studies presenting an imbalance in the mesopelagic carbon budget highlight that respiration exceeds the organic carbon supply (i.e. Burd et al., 2010).

p. 1, line 30: the b parameter in the Martin's curve is not the depth at which the material is remineralised.

p. 2, line 12: the "excess of POC supply" and "excess of respiration" compared to...?

p. 3, line 5: "missing carbon sink", again even if the reader understands what it is meant here, this is confusing as the missing term is an amount of organic matter needed to sustain observed respiration rates.

p. 3, line 15: why not calculating the mixed layer depth based on seawater density? Park et al. [1998] showed that the temperature alone might not be a reliable proxy for mixed layer calculations. Given the small variations of salinity with depth, I do not suspect any major change of the MLD calculations if based for instance on the 0.02 sigma-theta density difference criteria, but this could be checked easily.

p. 3, line 24: I doubt that bloom stage can be inferred that way. At most, Chl. a surface concentrations from satellite data can inform on biomass levels at a given time. Unless the bloom is considered as a whole, regardless of species successions and size classes – which should be avoided – the term 'bloom stage' denotes rather a moment in phytoplankton community successions and physiological state and can be estimated by sampling the plankton communities.

p. 4, line 1: considering that the height of the MSC is 1.53 m, it means that the particles collected are those which settled at approximately 18.4 m d-1 or faster. It seems that excluding all particles settling slower than this speed could have removed a non-negligible fraction of the flux, especially because of the findings of Riley et al. [2012] who deployed MSCs at the same site in summer 2009. They found that the

POC flux from fast-sinking particles was 54 mg C m-2 d-1, while the POC flux carried by slow-sinking particles (not sampled here) was 92 mg C m-2 d-1, suggesting that slow-sinking particles might sustain most of the POC flux. As noted above, it would have been necessary to also estimate the total POC flux collected in the MSC to have some idea of the contribution of the fast-sinking POC to the total flux.

p. 4, line 20: what sort of composition is assessed here? If based on the images showed in the supplementary material, only a very subjective idea of particle composition, and very likely not quantitative, can be accessed this way.

p. 4, line 25: how the decision was made to apply a given formula to each particle? Did the authors used morphological data from Image J (e.g. aspect ratio threshold to apply formulae for a cylinder or a sphere)?

p. 4, line 27: if an uncertainty is related to carbon content in FP then it should increase with depth where FP become predominant. It should be taken into account when trying to explain the unexpected increase of the POC flux between 203 and 500 m (section 3.4).

p. 5, line 24: "similar ESD" needs to be rephrased as if I well understood aggregates have been sorted in size classes not similar ESDs.

p. 5, line 26: "Where possible we measured POC...". This needs to be clarified.

p. 5, line 27: please change "fractal shape" to "fractal geometry".

p. 5, line 28: the correct reference is Alldredge [1998].

p. 6, line 1: the rotation speed of 3 rpm seems incredibly fast, and the incubation time of 7 days (p. 10, line 21) needed to obtain the first signs of aggregation surprisingly long. The speed of 3 rpm chosen by Iversen and Ploug (2010) was adapted to aggregates sinking much faster (due to ballast effect), and thus needing a very high rotation speed in order to keep them in suspension and avoid collision with the walls of the tank. I suspect that the same rotation speed used here was too fast to allow the particles to

settle at any point of their rotation around the centre of the tank, minimising aggregation by differential settling. It would explain why obtaining 'decent-size' aggregates have required a very long incubation of 9 days which is surely a factor that played in bacterial remineralisation. A very large fraction of the POC content in the aggregates may have been respired by the end of the incubation. Since no measure of POC, DOC and Dissolved Inorganic Carbon (DIC) was made at the beginning of the incubation, but only POC content measured at the end, no information is available on the solubilisation of the POC to DOC and its subsequent remineralisation to Dissolved Inorganic Carbon (DIC), rendering any conclusion impossible.

p. 6, line 13: this is another potential bias in the study since no evidence is given that zooplankton abundances and respiration rates measured in August 2009 by Giering et al. (2014) are representative of the system studied here. Maybe the authors can look for existing data of inter-annual variations of zooplankton abundances at the PAP site. Even if the importance of this term is minimised in the Section 4.5 and the authors estimate that it cannot close the carbon budget, it also needs to be calculated as accurately as possible (by direct measurements) as any biogeochemical budget needs a careful consideration of every term.

p. 7, line 9: please replace "grazing" by "coprophagy" as it is the correct term.

p. 7, lines 16-17: "PA sinking velocity was significantly ($R^2 = 0.163$, $p<0.0001$, $n=98$) correlated with ESD". It seems to me that these statistics suggest precisely the opposite here. It needs clarification. Also, the 6 outliers excluded from the relationship should be marked in Figure 3.

p. 7, line 22: correct reference is Laurenceau-Cornec et al. (2015).

p. 7, line 23: the size of aggregates formed in roller tanks is controlled by the time of incubation, initial cell concentration, stickiness, etc. and can hardly be compared between studies.

p. 8, lines 3-9: this paragraph seems to belong to the Discussion section rather than the Results.

p. 8, line 18: "Based on pool measurements...". This needs more details. In particular what were the size classes (and their width) used to pool the aggregates.

p. 8, line 21: this refers to the comment on p. 6, line 1, that a large fraction of the POC should have been respired during the prolonged incubation (as noted by the authors p. 10, lines 18-19). Also, the aggregation processes likely affected by tank rotation speed may have controlled the fractal dimension of the aggregates and influence their POC:Vol ratios. Maybe the authors can somehow estimate the fractal structure of the aggregates using their images following Kilps et al. [1994].

p. 8, line 31: this refers to the previous general comment that estimating bacteria abundance in the aggregates could have brought valuable insights here.

p. 8, line 33: please change "temperature are higher" to "temperature changes are higher".

p. 10, line 15: does it imply that the aggregates collected with the MSC have been subjected to fragmentation or aggregation subsequent to their sampling in the water column? If the data from the MSC are assumed to be valid then what was the motivation of using artificial roller tank aggregates known to represent poorly real particles?

p. 10, line 33: again, "aggregate composition" is a very vague description. A proper composition analysis should be chemical and/or taxonomic. More details on what was really assessed here are needed.

p. 11, line 11: I strongly disagree and think that a measure of bacterial abundance was in the scope of this study. The authors need to justify the absence of these measurements in other ways.

p. 11, line 22: why not trying to explain this unexpected (and thus interesting) value rather than excluding it from the study?

p. 12, line 27: "... which is likely to be less accessible to free-living microbes". This is a very important statement on which rely the justification that free-living bacterial respiration was not measured here. More details are needed on "less accessible". To what extent? Some additional quantitative informations supported by adequate references are needed.

COMMENTS ON FIGURES

Fig. 1. a: perhaps the authors could reduce the scale of this map and centre it on the PAP site as it is very difficult to see any mesoscale structure at this scale (if this is what is intended). Also, why not choose a satellite product which encompasses the sampling period of the study?

Fig. 3: A log scale on the Y-axis might improve the readability of the different sets of aggregates from each depth. Also, separate sinking velocity-size fits for each set of aggregates from distinct depth (one fit by color) could reveal interesting findings, especially if the structure and/or composition of the aggregates are assumed to vary with depth.

REFERENCES

Alldredge, A. (1998), The carbon, nitrogen and mass content of marine snow as a function of aggregate size, Deep Sea Research Part I: Oceanographic Research Papers, 45(4–5), 529 - 541, doi: 10.1016/S0967-0637(97)00048-4. Burd, A. B., et al. (2010), Assessing the apparent imbalance between geochemical and biochemical indicators of meso- and bathypelagic biological activity: What the @$âŹŕ! is wrong with present calculations of carbon budgets?, Deep Sea Research Part II: Topical Studies in Oceanography, 57(16), 1557 - 1571, doi:10.1016/j.dsr2.2010.02.022. Giering, S. L. C., et al. (2014), Reconciliation of the carbon budget in the ocean's twilight zone, Nature, 507(7493), 480–483, doi: 10.1038/nature13123. Jackson, G. A. (1994), Particle trajectories in a rotating cylinder: implications for aggregation incubations, Deep-Sea Research Part I: Oceanographic Research Papers, 41(3), 429–

437, doi:10.1016/0967-0637(94)90089-2. Kilps, J. R., B. E. Logan, and A. L. Alldredge (1994), Fractal dimensions of marine snow determined from image analysis of in situ photographs, Deep Sea Research Part I: Oceanographic Research Papers, 41(8), 1159-1169, doi:10.1016/0967-0637(94)90038-8. Park, Y.-H., E. Charriaud, D. R. Pino, and C. Jeandel (1998), Seasonal and interannual variability of the mixed layer properties and steric height at station KERFIX, southwest of Kerguelen, Journal of Marine Systems, 17(1–4), 571–586, doi:10.1016/S0924-7963(98)00065-7. Riley, J., R. Sanders, C. Marsay, F. Le Moigne, E. Achterberg, and A. Poulton (2012), The relative contribution of fast and slow sinking particles to ocean carbon export, Global Biogeochemical Cycles, 26, GB1026.

---

## Author Response (AR1)

Summary: The authors use a combination of emerging and established methods to address a major outstanding question in chemical oceanography; namely, the degree to which particle flux attenuation (i.e., the progressive remineralization of sinking marine particles with depth) is a result of respiration by attached bacterial communities, compared with various other biological and abiotic processes. The authors have a robust and very welcome dataset that has the potential to contribute significantly to a growing body of work in this area. Using shipboard measurements of respiration on particle material retrieved from depth, the authors conclude (in agreement with several other very recent studies) that direct respiration by particle-attached bacterial communities can explain only a fraction of the attenuation that is observed in the environment using sediment traps. (The relative importance of attached bacterial respiration appears to increase with depth compared with other processes.) The other major (and truly novel) finding of the paper – which is not presently addressed in the abstract, but should be – is the very apparent and mysterious difference in amenability to heterotrophic respiration between actual sinking marine particle material and artificial aggregates created in roller tanks. This is striking, because the latter have been invoked in dozens of experiments over the past 20 years as proxies for true sinking particle material. In the present study, the labilities of the two types of particles appear to differ very significantly.

General impression and recommendation: This is scientifically interesting work that deserves publication in Biogeosciences. The conclusions advance the dialog incremetally in one avenue of research (the partitioning of particle flux attenuation between attached respiration and other processes) and present a very new and striking finding with regard to the roller tank particles versus "real" particles. The authors' central findings and conclusions are acceptable and meet the criteria for publication in Biogeosciences. However, I have some very significant concerns which pertain primarily to (1) the authors' reading, interpretation, and presentation of the existing literature and state of the art as regards the biological pump (several issues), (2) the interpretation of respiration rates obtained using the authors' method, and (3) the structure of their manuscript. The authors could address these concerns with some re-analysis (particularly a more diligent consideration and assimilation of uncertainties), a more careful (and nuanced) interpretation of the data, to include mention in the manuscript of some very relevant caveats, and some re-writing of the manuscript, in places. In addition, important details of calculations and mathematical methods are omitted.

The manuscript is generally well-written and the figures are generally clear. I have some suggestions for the authors on two of their figures; addressing these should be trivial.

AC: Thank you very much for taking the time to review our manuscript and for the insightful comments. We have addressed your comments below which we think has greatly improved the manuscript. Please see our detailed response and attached marked up manuscript showing all amendments.

General comments:

(1)        In both the abstract and body of the manuscript, the authors generally overstate the novelty of their finding with regard to the importance of particle-attached respiration.

As they note on pp. 11-12, a small number of other recent studies have also presented similar results. I would urge the authors' to revisit the manuscript and ensure its tone reflects the findings in these existing studies apart from the citations on pp. 11-12.

AC: We agree that our work supports and builds upon and a number of recent studies and have tweaked the manuscript to ensure that we are not overstating the novelty of our work. For example in the introduction (review document page 33, line 31);

"To the best of our knowledge only two studies have combined direct measures of respiration on aggregates collected at depth with measurements of POC flux (Collins et al., 2015; McDonnell et al., 2015), both of which lack sufficient vertical resolution in the upper mesopelagic to capture the region of most rapid change. Collins et al. (2015) measure rates of substrate-specific microbial respiration of 0.007±0.003 d$^{-1}$ to 0.173±0.105 d$^{-1}$ in the North Atlantic, with rates of 0.01±0.02 d$^{-1}$ and 0.4±0.1 d$^{-1}$ measured by McDonnell et al. (2015) at the Western Antarctic Peninsula and Bermuda Atlantic Time Series respectively. Previous studies are therefore inconclusive as to the importance of particle associated microbes on the attenuation of POC, with some studies suggesting they play a minor role (Alldredge and Youngbluth, 1985; Collins et al., 2015; Ducklow et al., 1982; Karl et al., 1988) and others suggesting a larger contribution (Iversen and Ploug, 2013; Ploug et al., 1999; Turley and Stutt, 2000). Also on page 34 line 11:

"To build upon these previous studies we assess the role of particle associated microbial respiration in POC flux attenuation, presenting a vertical profile of particle associated respiration rates measured on individual marine snow particles collected at depth."

We also now note that we are building upon previous studies elsewhere in the discussion, such as in section 4.3 (review document page 45, line 4) and 4.4 (review document page 46, line 6):

"Despite the uncertainties in the mechanisms governing rates of particle associated microbial respiration, we are still able to assess the importance of particle associated microbial respiration on the attenuation of POC in the mesopelagic and compare our results to the small number of other recent studies (Collins et al., 2015; McDonnell et al., 2015)."

"Recent assessments of the mesopelagic carbon budget at the PAP site, although balanced over 50-1000 m, revealed an imbalance when the upper and lower mesopelagic were examined separately (Giering et al., 2014). Particle associated respiration was not directly measured in the aforementioned study, and hence we assess whether this term could help to explain observed imbalances. In this way we test whether the low respiration rates (0.001-0.173 d$^{-1}$) measured by Collins et al. (2015) are also applicable to our study site or whether the higher rates, such as observed in the western subtropical North Atlantic gyre (0.4 d$^{-1}$) by McDonnell et al. (2015) are more appropriate."

(2)          I have some significant concerns about the degree of handling, size-fractionation, and manipulation involved in the collection and incubation of the particle material in this study. First, the authors invoke the term in situ for these measurements; this is neither correct nor appropriate. Second, I am concerned that the particle material used for the incubation studies was manipulated and size-fractionated to a degree that makes comparison of respiration rates measured on the material to the MSC sediment trap flux measurements very questionable (specific concerns below).

AC: We have changed our terminology from in situ aggregates to natural aggregates. We have addressed these concerns about particle manipulation in your specific comment below. Briefly, the use of a wide bore pipette avoids fragmentation of the particles and as carbon-specific respiration rates are size independent, if any size fractionation occurred then this would not affect our conclusions based on carbon-specific respiration rates.

(3)          The authors' findings regarding the difference in lability between the two types of particles is really the novel contribution of this paper; I would encourage them to revisit the manuscript with an eye toward elevating the importance of these conclusions.

AC: We have added text into the discussion to highlight the differences we have found between artificial and natural aggregates. Unfortunately we do not have a chemical measure of lability, or a measure of microbial abundance on the different aggregate types and so can only speculate as to the reasons for the differences between them. Artificial aggregates were formed from waters collected at a shallower depth than collected natural aggregates, and hence we cannot determine if the differences in respiration rates between the two aggregates were a result of the roller tank

formation or simply due to their different depth of origin. For this reason we exercise caution when describing the potential importance of these conclusions. We suggest in the second paragraph of section 4.2 that this as an important area for future research, and recommend comparing the lability and microbial abundances on natural aggregates with artificial aggregates formed from plankton collected from the same depth as the natural aggregates.

(4)        The authors do not describe at all their methods for error assimilation and propagation; in fact, it is unclear whether this was even considered. As the authors are aware, the business of extrapolating rates of respiration on single particles in a highly controlled environment to the carbon cycle of a marine water column involves a large number of calculations involving a number of conversion and scaling factors, each of which have their own uncertainties. I would like to see some sort of more robust error propagation/analysis in the manuscript.

AC: In the previous version of the manuscript, when calculating average respiration rates ($C_{spec}$) for each depth horizon (which are used in subsequent calculations in the manuscript) we calculated upper and lower bounds based on the range of respiration rates measured in PA. Based on your recommendations we have now conducted a more thorough assessment of our uncertainties using a Monte Carlo approach (bootstrap analysis with 10,000 simulations). We have added details of this in section 2.6.

The calculated uncertainties are presented as error bars in Figure 9A. We believe, that despite uncertainties, our measurements make the most of available data (both our own and in the literature) to provide further insights into the role of particle associated microbial respiration on the attenuation of POC flux. We calculate that particle associated microbial respiration can account for only 1-14% of POC loss in the upper mesopelagic and hence the further error propagation we have carried out foes not change our conclusions. We have added the following to the manuscript (review document page 38, line 3):

"Time and methodological constraints of measuring very small particles, prohibited us from measuring the respiration rate of every particle collected in the MSC. Hence, before using these measurements to assess the contribution of particle associated microbial respiration to the mesopelagic carbon budget, we must first define upper and lower bounds to our estimates based on our uncertainties. We conduct a Monte Carlo analysis (with 10,000 iterations) of the individual parameters used in the calculations of carbon-specific respiration and remineralization length scale. We randomly sample (with replacement) our measured volumetric oxygen respiration rates at each depth. For each of these randomly selected particles we use the corresponding sinking velocity and ESD in subsequent calculations of carbon-specific respiration and remineralization length scale. For the RQ, used to convert $O_2$ to $CO_2$, we define a uniform distribution of possible values over the range of RQ values typically applied in the literature (0.7-1.2, Berggren et al., (2012)). PA were pooled into size classes and could only be measured once for POC content. For each depth, we create a normal distribution of possible POC:volume ratios for each size class, with our measured value as the mean, and standard deviation calculated from the standard deviation of the individual aggregate volumes within a size class. Based on the 10,000 iterations for each of the aforementioned parameters we obtain a range of estimates for the remineralization length scale (via particle associated microbial respiration) at each depth. We then use the mean of these distributions ±standard deviation to put error bounds on our estimates of the POC loss via particle associated microbial respiration."

Specific comments:

Abstract generally: Why no mention of the roller tank versus "real" particle respiration rate comparison?

AC: The most widely relevant part of our work is the low measured rates of particle associated microbial respiration and hence we have written the abstract to match this. The comparison with roller tank and in situ collected aggregates is not the main focus of our study here, partly because they were not conducted with water from the same depth (see comment above). The roller tank aggregates provide additional respiration rate estimates at a shallower depth than was sampled from the marine snow catcher, but it is difficult to make direct comparisons to the in situ collected particles because of these depth differences. We therefore do not stress these results in the abstract. See also response to general comment above.

p. 1, lines 11-12: Logic suggests the first sentence of the abstract should be restated in the converse, i.e., "Atmospheric levels of carbon dioxide are tightly linked to the depth at which sinking POC is remineralized in the ocean."

AC: Restructured sentence as suggested.

p. 1, lines 17-18: I am confused by the authors' statement of their hypothesis here. Perhaps they mean something more specific, i.e., "... the missing sink for particulate carbon in the upper mesopelagic"? In the growing number of recent studies in which other oceanographers have been unable to close mesopelagic carbon budgets (those in which both particulate and dissolved/suspended water column phases have been considered together), the problem has been one of an apparent undersupply of total carbon to the system, not a problem of too little respiration. However, if the authors are only considering the particulate phase, then this supposition would make more sense since the studies they reference on pp. 11-12 have demonstrated the opposite is true when considering the sinking particulate phase exclusively.

AC: We have altered the abstract to clarify that we are addressing the particulate phase only (review document page 32, line 23).
"We test the hypothesis that particle-attached microbes contribute significantly to community respiration in the mesopelagic, measuring particle associated microbial respiration of POC, through shipboard measurements on individual marine snow aggregates collected at depth. We find very low rates of both absolute and carbon-specific particle associated microbial respiration ($<3\%$ $d^{-1}$), suggesting that this term cannot solve imbalances in the upper mesopelagic POC budget."

p. 1, line 15: Confusing as currently stated. Try: "Attempts to balance POC supply to discrete depth layers of the mesopelagic..." (if that is in fact what the authors mean)

AC: Thank you for the good suggestion, the sentence has been altered.

p. 1, line 19 ff: Use of in situ confusing and inappropriate. If the authors mean here that the particles were collected from depth, then that should be stated; anything collected from the ocean in the course of oceanographic research can be said to have been collected "in situ." The respiration measurements the authors make this study are not in situ measurements; the only true in situ particle respiration measurements that I know of are those of McDonnell et al. (2015), obtained using the RESPIRE device. The measurements in the present study, as in Collins et al. (2015), are shipboard measurements made using material collected from depth.

AC: We agree with your judgement and have replaced in situ with natural aggregates collected at depth ($PA_n$), and use this throughout the manuscript.

p. 1, line 25: This shift could also be ultimately to DOC (not simply to suspended POC).

AC: The shift could indeed be to DOC also, but we attempt to account for losses to DOC via solubilization, and these alone do not seem to be able to address the mismatch. Previous studies including DOC still find imbalances and so although this may indeed be an ultimate fate, we focus here on processes that could account directly for the loss of POC.

p. 1, line 30: This is an incorrect interpretation of Martin's "b"; b does not represent the depth at which material is remineralized. (In the length-scale-based parameterization the authors invoke on p. 11, the interval z-z0 represents the depth interval over which material is remineralized.) This is one of the limitations of the Martin formulation, since the exponent doesn't really have any explicit meaning.

AC: Thank you for highlighting our mistake; we have removed the reference to Martin's b here.

p. 3, lines 3-4: I am not sure this is true; McDonnell et al. and Collins et al. both made profiles of particle-associated respiration rates using actual particle material. Is it the authors' contention that their use of "individual" particles makes this the first such study? Seems like a rather qualified claim to novelty that could be omitted from the paper without affecting its impact or conclusions.

AC: Rephrased as follows (review document page 34, line 11):

"To build upon these previous studies we assess the role of particle associated microbial respiration in POC flux attenuation, presenting a vertical profile of particle associated respiration rates measured on individual marine snow particles collected at depth."

p. 3, line 15: How did this measure of MLD compare to the 1% PAR level (traditionally invoked as the base of the euphotic zone, which is the reference depth for most other particle flux studies).

AC: We have added the following sentence to the results section (review document page 39, line 20):

"The MLD was typically within ±5 m of the 1% photosynthetically active radiation (PAR) level."

p. 4: I have some concerns about the representativeness of the respiration rates obtained from the authors' method, which appeared to require very extensive handling and manipulation in the selection and isolation of individual particles. Wouldn't this method of "plucking" by eye perhaps result in separation of the particles from the microbial communities with which they were likely associated in the environment? Further, how do the authors justify the assumption that the sum of the individual rates obtained on a few chosen particles can be applied to estimate rates of removal vis-a-vis the complete, heterogeneous collection of particles in the corresponding MSC fractions used for flux determination? Would it not have been better to measure rates on an assemblage of particles that was (a) subjected to less handling and (b) was more completely representative of the wider spectrum of shapes and sizes of the particles in the traps? I am not suggesting the errors introduced by this approach render it invalid (particularly given the enormous uncertainties involved in every other aspect of this sort of work), but a more complete discussion of the implications of their decision would be welcome.

AC: In this study we use a marine snow catcher as means to obtain natural aggregates. This is a closing water bottle, which is deployed to the depth of interest, closed trapping a parcel of water (95 L) then returned to the deck to settle. The settling period was kept short to two hours to minimize any potential "bottle-effect", following which the entire particle collector tray was moved to the temperature controlled laboratory to reduce handling. We have added a sentence to the methods to make clear that the MSC is a closing bottle not a sediment trap.

The process of removing the particles from the particle collector tray to the flow chamber using a wide bore pipette involves minimum disturbance of the particles and we did not observe any fragmentation of particles. We note here also that as carbon-specific respiration rates (as used for the estimation of removal rates) are independent of size (Fig 8), our calculations are not affected by any bias towards small or large particles. The following sentences have been added to the text (review document page 36, line 21):

"The wide bore pipette lifts the particles with the surrounding water so that the particles remain suspended in water during the handling and minimal physical stress is exerted on the particles. The microbial communities associated with the aggregates are not removed by this method (Kiørboe et al., 2002)."

The flow chamber is, we believe, one of the best methods available for respiration measurements, all of which have their limitations. By keeping the aggregates suspended, the flow chamber is able to mimic the settling of the aggregates since the flow velocity is adjusted to match the settling velocity of the aggregate. If the aggregate was placed on a solid surface oxygen would become diffusion limited, however, when allowing the aggregate to settle through oxygen saturated water we get very close to the conditions that the microbial communities within settling aggregates experience in situ (Ploug and Jorgensen, 1999). The flow chamber has been successfully used in a number of studies on marine aggregates (e.g. Iversen et al. 2010; Gärdes et al. 2011; Ploug & Bergkvist 2015). By suspending the aggregates in a flow, we are able to more accurately replicate natural conditions.

Addressing your second point about the summing of individual particles, we agree that ideally it would have been possible to measure all the particles in every sample but this was unfortunately not possible due to time and methodological constraints. However, as PA make up the bulk of the sinking material, >90% at depths 36-128 m, and still 67% at 500 m, we believe that our measurements accurately reflect the bulk of sinking material collected in the tray (as observed via microscope prior to handling). Even if respiration rates were high enough to remove all other types of

sinking POC, this would still not be able to explain the high loss rates of POC that we observed in the upper mesopelagic. We have added the following additional text in section 4.3 (review document page 45, line 19) to state the limitations with particle type.

"Our measurements are based on a sub sample of the total assemblage of particles found in the water column, in particular only PA. If rates of microbial respiration are vastly different on other particle types then this would affect our calculations of POC removal by particle associated microbes. However, considering the dominance of PA in our samples (Fig. 2), we believe our calculations reflect the bulk of the sinking material at the time of sampling. We were not able to measure respiration rates on FP due to their low numbers and small size which adds uncertainty to our estimate of the contribution of particle associated microbial respiration to POC loss. However, even if FP were respired completely, they account for less than 10% of the flux between 36-128 m and thus could not resolve the large imbalances between POC supply and respiration that we observe in the upper mesopelagic."

p. 5, line 25: Conversions of PA volume to what?

AC: Edited sentence (review document page 37, line 12) to clarify:

"This enabled the carbon content per unit volume (mg C mm$^{-3}$) for each size class at each depth range to be calculated and hence POC content of individual PA to be estimated."

p. 5, line 30: The way the authors have currently defined Cresp (line 18) and [POC] (line 29), it seems to me this equation yields a quantity Cspec with units of length (m) per unit time, not simply time-1. Perhaps the authors can clarify?

AC: Thank you for drawing our attention to this typo. We have amended the quoted units for Cresp, units are mg C mm$^{-3}$, which results in Cspec with units of d$^{-1}$.

p. 6, line 6: What is meant by "exponential fit" (i.e., what is the form of the equation)?

AC: Added text (review document page 37, line 28) to clarify form:

"…compared to an exponential fit ($R^2$=0.30, $p$=0.128, n=9) (of form, $F_z = F_0 \exp(z-z_0/z^*)$, as in Buesseler and Boyd (2009))."

p. 6, line 25: Are the 1.9 and 1.4 mg m-3 figures from discrete measurements or ocean color data?

AC: Added 'based on discrete measurements' to clarify.

p. 7, lines 7-10: Depending on the time of day at which the traps were collected, this increase in FP numbers could also be aliasing daily vertical migration.

AC: Yes, the timings are such that increased FP could be due to zooplankton migrating to depths >100m at sunrise, and these FP sinking further through the water column. We have added zooplankton diel vertical migration as a possible explanatory factor in this sentence (review document page 39, line 29).

"The increase in FP numbers below 100 m could be due to an increase in zooplankton populations with depth, zooplankton diel vertical migration, and/or increased FP loss in the upper mesopelagic due to processes such as fragmentation and coprophagy."

p. 9, lines 16-17: I do not understand the authors' meaning here. In addition, what is meant by "similarity"; this is a nonspecific and vague term.

AC: We have reworded these sentences to explain our meaning more clearly (review document page 42, line 17):

"Despite the consistency of average values of C$_{spec}$ at each depth horizon, there was large variability in C$_{spec}$ for individual aggregates within each depth horizon (full dataset range: 0.002-0.031 d$^{-1}$, Fig, 7). The calculated standard error was consistent across all depths (0.002—0.003) which implies that the factors driving the variability in C$_{spec}$ are unchanging with depth, suggesting controls on degradation are already determined at shallow depths"

p. 9, lines 19-20: Version of record of this paper is a 2015 date of publication.

AC: Corrected publication date

p. 9, last paragraph: Given the very significant influence of temperature on metabolism, why were the particles incubated at a static 10°C rather than at in situ temperature? If because only a limited number of incubators were available for large number of particles, this should be stated on p. 4.

AC: Shipboard considerations meant it was only possible to incubate at one temperature, and 10 °C was chosen as the minimum temperature likely to be encountered in the upper 500 m of the water column. We have stated this in section 2.4 as recommended.

"Only one incubation temperature was possible due to laboratory and space limitations."

p. 11, first full paragraph: This is the most intriguing and novel finding of the authors' study, and should be given more prominence.

AC: Thank you for your enthusiasm regarding these findings. We believe the most important aspect of our study to be the low rates of particle associated microbial respiration (see response to specific comment 1); nether the less the differences we find between natural and artificially made aggregates are indeed interesting. Due to several uncertainties, we are only able to make speculations about the possible differences in microbial abundance in these aggregate types but believe that this should be considered a priority for future research. Following your advice, we have added text in the manuscript, particularly at the end of section 4.2 (review document page 44, line 24) to increase the prominence of this finding as suggested but also explain our uncertainties in making strong comparisons.
"Over the past couple of decades there have been numerous studies utilizing roller tanks to create artificial aggregates from natural phytoplankton assemblages in an attempt to replicate natural sinking particulates, investigating processes such as aggregation, sinking velocity, ballasting and degradation (e.g. Iversen and Robert, 2015; Iversen et al., 2010; Laurenceau-Cornec et al., 2015; Shanks and Edmondson, 1989). In the present study, we sampled the natural phytoplankton assemblage from waters combining the highest POC concentration and chlorophyll fluorescence in an attempt to assess the aggregation potential in the most productive water strata. As $PA_r$ were formed from material at shallower depths than $PA_n$ were collected, we cannot be certain that the observed differences are not depth related. Further, we left the roller tanks to rotate for a period of 9 days in the dark and cold to simulate POC aggregation in the water column, accelerating plankton death and vulnerability to microbial degradation. These conditions deviate from natural conditions in which the plankton sinking from 12 m depth would have experienced progressive temperature and light decreases as well as changes in grazing pressure. Therefore, it is difficult to ascertain the cause of differences between roller tank formed and in situ collected aggregates, and rather we use the roller tank formed aggregates to get an estimate of the carbon-specific respiration rate of the aggregates at shallow depths".

p. 11, line 29: "Slow" compared to what? I would suggest the rates of respiration measured by others in water column samples are truly "slow" compared with particle- attached/associated rates.

AC: We agree that sinking particles are typically hotspots of activity in comparison to the surrounding water column, but suggest that zooplankton grazing and fragmentation leads to more rapid loss of large sinking particles. Fragmented material is likely ultimately respired by microbes, both particle associated and free-living, but we refer here to the direct loss of large sinking POC. We have amended the text as follows (review document page 45, line 30):

"Despite being hotspots for microbial activity compared to the water column (Thiele et al., 2015), particle associated microbial respiration may still be a minor contributor to the reduction in POC flux when compared to rapid loss via processes such as zooplankton grazing and fragmentation (Dilling and Alldredge, 2000; Stemmann et al., 2000; Svensen et al., 2014)."

p. 12, line 19: What are the error bounds on this 139.3 mg C m-2 d-1 figure?

AC: Following our additional uncertainty analysis we have recalculated our error bounds and quote them in the text (review document page 46, line 26):

"…excess POC supply of 118 mg C m$^2$ d$^{-1}$ (60-177 mg C m$^{-2}$ d$^{-1}$ considering our calculated error bounds on rates of particle

associated microbial respiration and POC flux measurements) over the upper 36-200 m."

p. 13, first paragraph: The authors do not mention the additional possible mechanical processes of shear or turbulent disaggregation.

AC: Thank you for drawing this oversight to our attention. We have added the following paragraph in section 4.5 (review document page 48, line 18) where we address the possible missing parts of the budget:

"Additionally we are not able to measure mechanical disaggregation via processes such as fluid shear which could provide additional losses of large sinking POC. The forces required to break apart large marine snow aggregates have been shown to be higher than typical estimates of energy dissipation in the ocean, suggesting that this would not be a major loss process (Alldredge et al., 1990). Physical disaggregation could be more important in surface waters where dissipation rates can exceed the forces required to break marine snow aggregates (Alldredge et al., 1990; Burd and Jackson, 2009). However, we suspect that only a small fraction of sinking POC would be fragmented by abiotic processes to particles <0.15mm and hence would not explain the large loss of fast sinking POC measured in this study."

p. 13, line 33: I concur wholeheartedly! The authors would do well to make a connection here to very recent and compelling work by Biard et al. (2016), showing the ocean may contain large numbers of these protists.

AC: Thank you for drawing our attention to this interesting paper. We have added a reference to it to support our statement (review document page 48, line 25).

"In order to address the imbalances in the sources and sinks of fast sinking POC to the upper mesopelagic we require an additional loss process of POC. One key term missing from the budget is that of free-living protozoans which would not be collected in zooplankton nets and can make up a substantial part of marine planktonic ecosystems (Biard et al., 2016)".

pp. 14-15: I commend the authors for their thoughtful and excellent discussion here.

p. 14, line 30: Use of in situ misleading and redundant.

AC: Changed to 'at depth'.

Comments on figures and tables:

Figure 1: Panel (c) might be improved by the superimposition of some chlorophyll concentrations measured concurrently in the water samples.

AC: Added points of chlorophyll from discrete measurements from CTD as recommended to figure 1C and also added profile of chlorophyll in figure 1B.

Figure 2: From inspection of the figure, it seems to me the authors' decision to use only PA-derived rates (and not also FP-derived rates) is valid only for depths < 113 m. Below this depth, FP constitute a very significant fraction of the POC.

AC: We only measure respiration rates in PA due to the low numbers and small size of collected FP. We acknowledge that this increases the uncertainty in our predicted POC loss via particle associated microbes, particularly at depth where FP are more important. We have added text to section 4.3 (review document page 45, line 23 of the manuscript to acknowledge this limitation.

"We were not able to measure respiration rates on FP due to their low numbers and small size which adds uncertainty to our estimate of the contribution of particle associated microbial respiration to POC loss. However, even if FP were respired at significantly different rates, they account for less than 10% of the flux between 36-128 m and thus could not resolve the large imbalances between POC supply and respiration that we observe in the upper mesopelagic."

Figure 3: An excellent figure. Very clearly and compelling presents the most interesting conclusion of the authors' manuscript. However, could the authors provide the $R^2$ for these fits in addition to the p-values?

AC: We have added in $R^2$ values to the plot

Figure 4 (and corresponding presentation of this curve fit in the manuscript): What are the error bounds on the fitted parameters?

AC: To improve our error assessment of the b value obtained during this curve fit we have applied a bootstrap analysis with 100,000 simulations and quote the mean b+/- the standard deviation. This is stated in section 3.4 (review document page 41, line 2):

"To assess the uncertainty surrounding our calculated *b* value, we have applied a bootstrap analysis with 100,000 simulations giving a mean *b* of 0.67 ± 0.34 (standard deviation)."

Figures 6, 7: Use of "in situ" to characterize the rates presented in these figures is particularly misleading.

AC: We have amended the figures to say "natural" rather than in situ.

Figure 9: In line with my concern regarding the compatibility of the authors' respiration rates with their trap flux data: Is the "aggregate POC" represented by the black bars the same POC as that supplied in the incubations used to determine the respiration rates? Seems to me this is not the case. I am not sure these can be directly compared in such a way. I am not sure I understand the meaning of the error bars ("upper" and "lower" estimates; are these error bounds obtained from some sort of uncertainty analysis?). Specifically, how were the error bars on the solubilization rates obtained, since these were based on a very hypothetical assumption?

AC: The POC flux shown in Figure 9 includes all types of fast sinking POC collected by the marine snow catcher and we have amended to legend to make sure it matches this. Please see our earlier comment above where we justify the application of our respiration rates measured on PA to the whole flux based on their dominance in our samples. We have amended the text within the figure caption to better explain and refer to the methods employed for error bar calculation. As was previously in the figure caption, these did not include an error estimate of solubilization due to our inability to constrain it. We have now removed the error bars from the POC sinks in figure 9b) to avoid misleading the reader as to the accuracy of these calculations.

[revised manuscript text omitted]

* * *
Biogeosciences Discuss., doi:10.5194/bg-2016-130-RC2, 2016

[Figure]

This manuscript tackles a major and current question in the understanding of biogeochemical cycles in the ocean regarding the processes which affect the remineralisation length scale in the twilight zone and influencing the efficiency of carbon sequestration in the deep ocean. More specifically, the authors address the question of the relative influence of particle-attached bacterial remineralisation to other processes responsible for Particulate Organic Carbon (POC) flux attenuation in the upper mesopelagic zone. A large range a methodologies are used and combined to test their main hypothesis that the loss of organic carbon due to bacterial communities attached to 'fast-sinking particles' is the missing term that may help to close the carbon budget in the mesopelagic. Most of the methods used are recent but not novel and proved to be robust in previous published studies. The choice of the study site located in the Porcupine Abyssal Plain (PAP) in the North Atlantic is motivated by an extensive and recent pre-existing literature in this area. The resulting dataset is certainly valuable and could contribute to the scientific community. Results of their measurements combined with numerous calculations – and assumptions where needed – suggest that particle-associated bacterial respiration is not sufficient to explain the missing loss of organic carbon in the upper mesopelagic. As an alternative hypothesis, the authors propose that organic carbon losses due to the fragmentation of large fast-sinking particles into smaller slow-sinking/suspended particles operated by zooplankton in the mesopelagic could be the main process to account for to explain the imbalances observed in the mesopelagic carbon budget. Apart from a few sections, the manuscript is well-written and easy to read. The figures are clear and well presented, but in some places modifications are needed to improve the messages intended.

**GENERAL COMMENTS AND RECOMMENDATION**

Overall, this manuscript leaves a good impression on the goals targeted and the amount of work achieved. However, there is a striking mismatch between the hypothesis tested, the type of measurements conducted and the conclusions developed. I acknowledge the large number of measurements and understand the challenge presented by onboard analyses and that compromises have always to be made in the choice of the parameters measured but the conclusions drawn in a study have to align with the data acquired and it is unfortunately not the case here. In order to properly test the relative importance of the respiration associated with particle-attached bacterial communities, a complete set of the other known processes responsible for organic carbon loss, along with a comprehensive inventory of carbon sources should have been measured. This work lacks measurements of major parameters essential to establish a valid carbon budget in the mesopelagic. As stated in the title of the last section of the Discussion ("The missing piece of the mesopelagic budget"), the authors make an attempt to find a missing piece of a puzzle which seems to already miss several other very important pieces.

AC: Dear reviewer, thank you very much for the thorough review. We are pleased that you generally liked our manuscript and hope that the changes in the revised manuscript now reconcile the hypothesis with the

measurements.

In particular,

(1)	the values of zooplankton abundance and respiration rates used in the calculations are inferred from another study carried out in the same area but 6 years before. To justify the validity of applying these external data to the present work, the authors state that sampling was conducted in both studies at the same stage of the seasonal cycle but no evidence are given to support it. For instance a measure of phytoplankton cell physiological state (Fv/Fm), phytoplankton community structure (often displaying the same successions each year), nutrient levels potentially indicating exhaustion, and zooplankton community structure could have provided some element of response.

AC: We agree with your concerns about the use of past data and have been able to source zooplankton net data taken during the 2015 research cruise through further collaborations with other scientists on board the ship. We now estimate zooplankton respiration from net tows carried out from 0-200 m during the cruise. Although we lose resolution by using these data, we believe that these data provide a more reliable assessment of POC loss processes at the time of our study. The zooplankton abundance and respiration estimates from this study (daytime: 0.3 g DW $m^{-2}$; night-time: 0.4-0.9 g DW $m^{-2}$; respiration: 5-10 mg C $m^{-2}$ $d^{-1}$ between 0-200 m) are in good agreement with the data collected 6 years ago (daytime: 0.2-0.4 g DW $m^{-2}$; night-time: 0.6-1.3 g DW $m^{-2}$; respiration: 10-18 mg C $m^{-2}$ $d^{-1}$ between 0-200 m). Respiration rates 6 years ago were slightly higher during one of their stations due to high abundance of amphipods and euphausiids. Nevertheless, the overall conclusion in our original submission has not been impacted, as zooplankton respiration is still only a small fraction of the POC loss over that depth horizon (~220 mg C $m^{-2}$ $d^{-1}$ between 0-200 m).

(2)	no estimation of the respiration due to free-living bacterias in the water column is made. Previous studies noted however that it may be the predominant term in total bacterial respiration (see Extended Data Figure 6, Giering et al., 2014).

AC: Although indeed free-living bacteria can make up a large proportion of total bacterial respiration, this requires conversion of large sinking particles to dissolved organic matter (Cho and Azam, 1988; Karl et al., 1988). In contrast to Giering et al. (2014), we focus only on the direct loss of POC, investigating the balances between sources and sinks in this carbon pool, and hence we do not include free-living bacterial respiration. Free-living respiration is typically measured via leucine incorporation, and requires two conversion factors for which there is a large range of literature estimates (see methods of Giering et al., 2014). Leucine conversion factors and prokaryotic growth efficiencies range by 2 orders of magnitude and hence are, we believe, associated with too high a degree of uncertainty to provide a useful comparison. These large uncertainties motivated our focus on only the POC pool of the carbon budget and we assess the direct loss via particle associated microbial respiration only.

(3)	the measurements of bacteria abundance in aggregates is also missing although it would have bring essential information to conclude on the observed variations of POC content and calculated specific respiration rates in the marine snow aggregates.

AC: We agree that information on bacterial abundance, and diversity would indeed be interesting and may help explain observed variability in our measured respiration rates. However, to obtain carbon-specific respiration rates it was necessary to combust the particles to obtain a POC measurement and therefore it was not possible to also obtain prokaryotic abundance for these aggregates. Moreover, in order to obtain the most accurate estimate of carbon flux from fast sinking particles, we chose to dedicate the rest of the particles to carbon flux measurement. However, during the cruise a microbiology team performed parallel MSC deployments devoted to linking aggregate carbon content with microbial abundance and activity. The analysis for these data is ongoing and will be the subject of a separate future paper by this team.

(4)	Only the 'fast-sinking' fraction of the particle flux is analysed. This is a major flaw in the study. While the 'slow-sinking' fraction might poorly contribute to the deep carbon export as noted by Riley et al. [2012], it is precisely because most of this fraction of the flux appears to be remineralised in the mesopelagic zone. It is very surprising to me that no total POC measurements from the MSC collection have been done to allow a comparison with POC contents measured in the aggregates selected for the analysis.

AC: See response to specific comment (p. 4 line 1) below where we state that slow sinking fluxes were <10% of the total flux and therefore were only a small component of the total POC flux. In addition rates of respiration on slow sinking particles have been shown to be different to those on fast sinking particles (Cavan et al. in review), and hence it would not be appropriate to apply our measured respiration rates to the slow sinking POC pool also.

(5)          no measurements of DOC was made even if it represents a non-negligible source of carbon to depth and surely needs to be accounted for in any carbon budget. Having this term would have informed – and possibly confirmed– some of the numerous  hypotheses postulated in the Discussion section.

AC: As note above, we focus only on the fast sinking POC part of the budget to avoid the introduction of additional uncertainties. We do not attempt to balance the whole carbon budget and have made modifications throughout the manuscript to make this clear and avoid confusion as to our intention. Assessing the balance of the whole carbon budget would indeed require measurements of DOC, however the source and sink terms of POC must also balance, enabling us to assess whether particle associated microbial respiration can help close this part of the full carbon budget. We believe that this is a useful step that will contribute to the wider scientific community and that the large number of carbon pools and interacting processes necessitate this step by step approach to make advances in the field. We must first balance the sources and sinks of POC before we can properly assess the transfer to DOC and subsequent sinks.

In a very honest approach, the authors detail all these missing terms in the last section of the Discussion and try to weigh their potential influence based on the literature. Unfortunately, no valid conclusion can be drawn based on so many assumptions and by using values which are themselves subjected to very large uncertainties and potentially not applicable in this system. As a result, the main conclusion – in fact an alternative  hypothesis – is disappointing since it echoes the conclusion made by Giering et al. [2014], but without bringing any additional piece of evidence to confirm it. Despite all these caveats, I believe that this work is worthy of being published, mostly for its very  valuable dataset. However, a complete restructuring of the manuscript is required. The  formulation of new hypotheses more in-line with the measurements conducted should  help in this difficult task. The objectives initially aimed in this work (i.e. to close the  carbon budget in the mesopelagic zone) have to be downgraded, but it should not minimise the importance of the findings centred on the variations of particle-associated  bacterial respiration as a function of depth.  As already noted in the title, I suggest to  present this work as a focus on this single parameter of importance: the depth-resolved  particle-associated microbial respiration in the northeast Atlantic. I recommend to highlight the variations of respiration observed at each depth and in each set of aggregates  from the same depth. A large fraction of the discussion is already framed around this aspect, but a thorough evaluation of the variability attributable to errors inherent to each measurement should be examined before trying to explain the potential real variability. It might appear that propagation of the uncertainties could alone explain the variability observed at a given depth, allowing then to fully explore the depth-related variability. Finally, I wonder if the roller tank aggregation experiment is a valuable addition to this work or at the opposite if it weakens the study by showing results that contradict those made in the water column. Roller tank-made aggregates have been excluded a long time ago as models to quantify processes in real particles [Jackson, 1994]. In addition, the choice of settings used in the roller tank experiment and the methods conducted are highly questionable (see specific comments), and any evidence based on these results should be taken very carefully.

AC: We thank the reviewer for taking the time to thoroughly review our manuscript and for the very useful insights and suggestions. We have amended the manuscript to clarify our intentions to assess the contribution of particle associated microbes to the loss of POC from the manuscript. See below our specific responses to the comments raised.

Following the suggestions of reviewer number 1, we have carried out more in depth uncertainty analysis, the results of which do not affect our conclusions. We are therefore confident that the addition of particle associated microbial respiration to the mesopelagic carbon budget, does not improve the balance. The purpose of out paper is not to try and create a mesopelagic carbon budget, but to assess an additional loss term and determine if this could explain some of the depth resolved mismatches in POC sources and sinks. We have changed the title of section 4.5 to "The missing piece of the mesopelagic carbon budget?", to better reflect that we are exploring the possible solutions to the current imbalances, rather than making a definitive budget.

We have edited the final paragraph of 4.4 (review document page 47, line 17) to make it clear that we are not suggesting

that we are attempting to balance the budget in this study.

"POC loss via zooplankton respiration, particle associated microbial respiration and solubilization, as typically invoked in model studies (e.g. Anderson and Tang, 2010) can therefore not account for observed losses of POC in the upper mesopelagic, suggesting our knowledge of the mesopelagic carbon budget is still poorly constrained and/or incomplete."

SPECIFIC COMMENTS

Abstract, line 16: "...with respiration sometimes being 50% lower than apparent carbon loss in the upper mesopelagic." This sentence is confusing and possibly incorrect. Did the authors mean 50% "higher" rather than "lower"? Most of the studies presenting an imbalance in the mesopelagic carbon budget highlight that respiration exceeds the organic carbon supply (i.e. Burd et al., 2010).

AC: We have restructured the abstract and removed this confusing sentence.

p. 1, line 30: the b parameter in the Martin's curve is not the depth at which the material is remineralised.

AC: Thank you for highlighting our mistake; we have removed the reference to Martin's b here.

p. 2, line 12: the "excess of POC supply" and "excess of respiration" compared to...?

AC: Text amended to clarify we are comparing sources and sinks (review document page 33, line 16):

"However, this study contained large and compensating imbalances between sources and sinks in upper and lower mesopelagic layers, with an excess of POC supply to the upper mesopelagic (50-150 m depth), and an excess of respiration in the lower mesopelagic (150-1000 m depth)."

p. 3, line 5: "missing carbon sink", again even if the reader understands what it is meant here, this is confusing as the missing term is an amount of organic matter needed to sustain observed respiration rates.

AC: We have amended the text here to make it clear that we are not trying to close the mesopelagic carbon budget. Instead we focus on the balance between sources and sinks of fast sinking POC, and assess whether our new measurements of particle associated microbial respiration can help improve this balance (review document page 34, line 13):

"In an attempt to assess whether this term can improve our ability to balance the fast sinking POC budget in the upper mesopelagic…"

p. 3, line 15: why not calculating the mixed layer depth based on seawater density? Park et al. [1998] showed that the temperature alone might not be a reliable proxy for mixed layer calculations. Given the small variations of salinity with depth, I do not suspect any major change of the MLD calculations if based for instance on the 0.02 sigma-theta density difference criteria, but this could be checked easily.

AC: Following your advice we have calculated the mixed layer depth (MLD) based on a range of density criterion 0.02-0.125 $\sigma_\theta$ and find that the criterion of 0.125 (Levitus, 1982) best captures the pycnocline depths observed in our study. The depths obtained are within ±5 m of the depths we calculate based on a temperature criterion and so we stick with our original definition of MLD.

p. 3, line 24: I doubt that bloom stage can be inferred that way. At most, Chl. a surface concentrations from satellite data can inform on biomass levels at a given time. Unless the bloom is considered as a whole, regardless of species successions and size classes – which should be avoided – the term 'bloom stage' denotes rather a moment in phytoplankton community successions and physiological state and can be estimated by sampling the plankton communities.

AC: Thank you for highlighting our error in terminology here, we have corrected the sentence to better explain our intended use of the satellite data (review document page 35, line 3).

"We examine changes in surface chlorophyll prior to and post sampling by averaging chlorophyll data over the study region (48.5-49.5 °N, 16.0-17.0 °W)."

p. 4, line 1: considering that the height of the MSC is 1.53 m, it means that the particles collected are those which settled at approximately 18.4 m d-1 or faster. It seems that excluding all particles settling slower than this speed could have removed a non-negligible fraction of the flux, especially because of the findings of Riley et al. [2012] who deployed MSCs at the same site in summer 2009. They found that the POC flux from fast-sinking particles was 54 mg C m-2 d-1, while the POC flux carried by slow-sinking particles (not sampled here) was 92 mg C m-2 d-1, suggesting that slow-sinking particles might sustain most of the POC flux. As noted above, it would have been necessary to also estimate the total POC flux collected in the MSC to have some idea of the contribution of the fast-sinking POC to the total flux.

AC: We made measurements of the slow sinking pool and found in contrast to Riley et al., (2012) that it only represented a small part of the total carbon flux (on average <10%). We therefore focus our study on only the loss of fast sinking POC as it is these particles that we measure respiration rates on. Recent work by Cavan et al., (in review) found significantly different rates of microbial respiration in fast and slow sinking carbon pools and therefore it would not be valid to apply our measured respiration rates to the slow sinking carbon pool. We have added the following additional text to the methods section (review document page 35, line 19):

"Slow sinking particles were also collected as of Riley et al., (2012), with slow sinking particle velocities calculated using the SETCOL method (Bienfang, 1981). Slow sinking fluxes were only a small part of the total sinking flux (on average <10%) and due to their slower sinking rate these particles do not penetrate as deeply in the mesopelagic. Hence we focus our study on fast sinking particles only."

p. 4, line 20: what sort of composition is assessed here? If based on the images showed in the supplementary material, only a very subjective idea of particle composition, and very likely not quantitative, can be accessed this way.

AC: Thank you for drawing our attention to this, we have replaced 'composition' with 'type' to clarify that we classified the particles into types as described in the text but did not assess composition (review document page 36, line 4).

"The type of fast sinking particles at each depth was assessed under a microscope and photographs taken using a Leica DM-IRB inverted microscope and Canon EOS 1100D camera. Particles were classified into phytodetrital aggregates (aggregations >0.15 mm equivalent spherical diameter (ESD) containing phytoplankton cells and other phytodetrital material, herein referred to as PA), faecal pellets (FP), and unidentified phytodetritus."

p. 4, line 25: how the decision was made to apply a given formula to each particle? Did the authors used morphological data from Image J (e.g. aspect ratio threshold to apply formulae for a cylinder or a sphere)?

AC: Classifications were done manually by A.Belcher based on the particle appearance. The morphologies were distinct allowing confident classification.

p. 4, line 27: if an uncertainty is related to carbon content in FP then it should increase with depth where FP become predominant. It should be taken into account when trying to explain the unexpected increase of the POC flux between 203 and 500 m (section 3.4).

AC: Yes you are correct in that the uncertainty in carbon content of FP would affect our estimates of the % contribution of each particle type to the total POC (Figure 2). We only use area derived POC from photos and conversions to get an estimate of the contribution of each particle type to the total mass of sinking material. These conversions are not used for the calculation of total POC flux as presented in Figure 4. The POC flux is based on direct measurement of POC (a split of the particles were placed on GF/F filters, see section 2.3). The increase in POC at 500 m cannot therefore be explained by uncertainties in literature values as no literature values were used in calculation of these fluxes.

p. 5, line 24: "similar ESD" needs to be rephrased as if I well understood aggregates have been sorted in size classes not similar ESDs.

AC: Sentence amended to:

"…before pooling PA into size classes based on ESD and placing onto pre-combusted…"

p. 5, line 26: "Where possible we measured POC...". This needs to be clarified.

AC: Clarified as follows (review document page 37, line 14):

"We measured POC to volume ratios of two sizes classes (typically <0.6 mm ESD, >0.6 mm ESD) at each depth horizon (with the exception of samples at 128, 200 and 500m where all measured particles were <0.6 mm ESD so only one size class was used)…"

p. 5, line 27: please change "fractal shape" to "fractal geometry".

AC: Amended

p. 5, line 28: the correct reference is Alldredge [1998].

AC: Amended

p. 6, line 1: the rotation speed of 3 rpm seems incredibly fast, and the incubation time of 7 days (p. 10, line 21) needed to obtain the first signs of aggregation surprisingly long. The speed of 3 rpm chosen by Iversen and Ploug (2010) was adapted to aggregates sinking much faster (due to ballast effect), and thus needing a very high rotation speed in order to keep them in suspension and avoid collision with the walls of the tank. I suspect that the same rotation speed used here was too fast to allow the particles to settle at any point of their rotation around the centre of the tank, minimising aggregation by differential settling. It would explain why obtaining 'decent-size' aggregates have required a very long incubation of 9 days which is surely a factor that played in bacterial remineralisation. A very large fraction of the POC content in the aggregates may have been respired by the end of the incubation. Since no measure of POC, DOC and Dissolved Inorganic Carbon (DIC) was made at the beginning of the incubation, but only POC content measured at the end, no information is available on the solubilisation of the POC to DOC and its subsequent remineralisation to Dissolved Inorganic Carbon (DIC), rendering any conclusion impossible.

AC: Thank you for these detailed and insightful comments. Indeed if the tank rotation speed is too fast then there is little time for the particles to settle which would reduce the rate of aggregation. We cannot be sure if the rotation speed had some inhibitory effect on the speed of aggregation but we observed particle formation after two days and aggregates increased in size during the incubation period. Additionally, a large number of the aggregates formed in the study of Iversen and Ploug (2010) fall in the range of sinking velocities that we measure on our roller tank formed aggregates (50-150 m d$^{-1}$), suggesting that aggregation processes are not inhibited at this speed. However, the long incubation time does increase the likelihood of "bottle-effects", and it could be that bacterial abundances were increased artificially high over the incubation period. However, long incubations (380 hr) were carried out by Iversen and Robert (2015) and no significant changes in carbon-specific respiration rates were observed in this time. Similarly Iversen and Ploug, (2013) carried out roller tank incubations at 4 and 15 °C over a period of a few weeks and did not observe any significant trend in carbon-specific respiration rates. These previous studies suggest that our incubation time of 9 days should not have influenced carbon-specific respiration rates. We have added the following sentence to section 4.2 (review document page 44, line 1):

"Although the length of the roller tank incubation may have allowed for a decrease in PA POC via microbial remineralisation, previous studies carrying out longer incubations (Iversen and Ploug, 2013; Iversen and Robert, 2015) do not measure significant changes of $C_{spec}$ over time suggesting that the long incubation would not bias rates of $C_{spec}$".

p. 6, line 13: this is another potential bias in the study since no evidence is given that zooplankton abundances and respiration rates measured in August 2009 by Giering et al. (2014) are representative of the system studied here. Maybe the authors can look for existing data of inter-annual variations of zooplankton abundances at the PAP site. Even if the importance of this term is minimised in the Section 4.5 and the authors estimate that it cannot close the carbon budget, it also needs to be calculated as accurately as possible (by direct measurements) as any biogeochemical budget needs a careful consideration of every term.

AC: We have added new zooplankton data from vertical net tows carried out during our research cruise in order to remove this potential bias (see sections 2.7, and 3.5). As these net data are integrated over 0-200 m we are only able to assess their

role in the loss of POC over this region. The zooplankton abundance and respiration estimates from this study (daytime: 0.3 g DW m$^{-2}$; night-time: 0.4-0.9 g DW m$^{-2}$; respiration: 5-10 mg C m$^{-2}$ d$^{-1}$ between 0-200 m) are in good agreement with the data collected 6 years ago (daytime: 0.2-0.4 g DW m$^{-2}$; night-time: 0.6-1.3 g DW m$^{-2}$; respiration: 10-18 mg C m$^{-2}$ d$^{-1}$ between 0-200 m). Respiration rates 6 years ago were slightly higher during one of their stations due to high abundance of amphipods and euphausiids. Nevertheless, the overall conclusion in our original submission has not been impacted, as zooplankton respiration is still only a small fraction of the POC loss over that depth horizon (~220 mg C m$^{-2}$ d$^{-1}$ between 0-200 m).

p. 7, line 9: please replace "grazing" by "coprophagy" as it is the correct term.

AC: Thank you for spotting our mistake. This has been corrected.

p. 7, lines 16-17: "PA sinking velocity was significantly (R2 = 0.163, p<0.0001, n=98) correlated with ESD". It seems to me that these statistics suggest precisely the opposite here. It needs clarification. Also, the 6 outliers excluded from the relationship should be marked in Figure 3.

AC: The statistics quoted here show that there is a significant relationship between sinking velocity and ESD with the low $R^2$ describing that there is large variability surrounding this relationship. To make this variability clear we have added the following sentence (review document page 40, line 10):

"The low $R^2$ shows that there is variability around this relationship, suggesting some heterogeneity in PA composition."

We have also added the outliers to the relationship to Figure 3.

p. 7, line 22: correct reference is Laurenceau-Cornec et al. (2015).

AC: Corrected

p. 7, line 23: the size of aggregates formed in roller tanks is controlled by the time of incubation, initial cell concentration, stickiness, etc. and can hardly be compared between studies.
AC: Thank you for picking up on this, we intended to only compare sinking velocities and have restructured the sentence. Although particle sinking velocities are also affected by many factors, are best compared for particles of similar sizes which is why we compare to only a subset of particles in the aforementioned study and quote the size range of this subset. We have restructured the sentence to make this clearer (review document page 40, line 15).
"A study in the Southern Ocean by Laurenceau-Cornec et al. (2015) also found size-specific (1.3-3.1 mm ESD) sinking velocities similar to those measured here (50-149 m d$^{-1}$)."
p. 8, lines 3-9: this paragraph seems to belong to the Discussion section rather than the Results.
AC: Although we do attempt to explain results here in the manner of a discussion, we have chosen to do this within the results section to keep the manuscript as concise as possible. We briefly explain the possible scenarios leading to an increase in flux with depth, but quickly move on to the main focus of our paper, the role of particle associated microbial respiration. An additional section would be needed in the discussion to discuss these results which we believe would unnecessarily increase the length of the article and break up the flow of the article.
p. 8, line 18: "Based on pool measurements...". This needs more details. In particular what were the size classes (and their width) used to pool the aggregates.
AC: We have altered the sentence for clarity and have referred the reader to section 2.4 where the separation into two size classes (<0.6, >0.6 mm ESD) is described (review document page 41, line 12).
"For each depth horizon we measure the POC contents of the PA used in respiration experiments (see section 2.4). We find POC contents of PA$_n$ ranging…"

p. 8, line 21: this refers to the comment on p. 6, line 1, that a large fraction of the POC should have been respired during the prolonged incubation (as noted by the authors p. 10, lines 18-19). Also, the aggregation processes likely affected by tank rotation speed may have controlled the fractal dimension of the aggregates and influence their POC:Vol ratios. Maybe the authors can somehow estimate the fractal structure of the aggregates using their images following Kilps et al. [1994].

AC: Although calculation of the fractal dimension would indeed provide another parameter to compare the roller tank aggregates with the in situ collected aggregates, as we do not have stacked images of each particle we think that the uncertainties in calculating the exact particle perimeter (as required for the Kilps et al. (1994)) are too high. Logan and Wilkinson (1990) derive a method using settling velocity versus size to estimate fractal dimension, however this too relies on a number of assumptions which we believe would increase uncertainty in our analysis and make the comparison less useful.

p. 8, line 31: this refers to the previous general comment that estimating bacteria abundance in the aggregates could have brought valuable insights here.

AC: See response to general comment

p. 8, line 33: please change "temperature are higher" to "temperature changes are higher".

AC: Changed

p. 10, line 15: does it imply that the aggregates collected with the MSC have been subjected to fragmentation or aggregation subsequent to their sampling in the water column? If the data from the MSC are assumed to be valid then what was the motivation of using artificial roller tank aggregates known to represent poorly real particles?

AC: We have amended this sentence to clarify our meaning here. Although a number of studies have noted some of the issues with roller tank studies, many studies still utilize roller tanks to collect particles, We therefore thought that we could make additional useful comparisons of respiration rates. We use the roller tank formed aggregates to provide data in the fluorescence maximum where we did not have a MSC deployment. The following text has been added in section 4.2 (review document page 44, line 24).

"Over the past couple of decades there have been numerous studies utilizing roller tanks to create artificial aggregates from natural phytoplankton assemblages in an attempt to replicate natural sinking particulates, investigating processes such as aggregation, sinking velocity, ballasting and degradation (e.g. Iversen and Robert, 2015; Iversen et al., 2010; Laurenceau-Cornec et al., 2015; Shanks and Edmondson, 1989). In the present study, we sampled the natural phytoplankton assemblage from waters combining the highest POC concentration and chlorophyll fluorescence in an attempt to assess the aggregation potential in the most productive water strata."

p. 10, line 33: again, "aggregate composition" is a very vague description. A proper composition analysis should be chemical and/or taxonomic. More details on what was really assessed here are needed.

AC: We have amended the sentence to clarify that only visual assessment of taxonomic composition was carried out and as already stated in the manuscript, we note that further chemical analyses would be needed to allow quantification of any differences between aggregates (review document page 44, line 8).

"However, we could not visually distinguish any clear differences in taxonomic composition of $PA_n$ collected at depth and roller tank PA based on SEM or light microscope imagery (supplementary Fig. S1)."

p. 11, line 11: I strongly disagree and think that a measure of bacterial abundance was in the scope of this study. The authors need to justify the absence of these measurements in other ways.

AC: We have amended the sentence to say that we were unable to measure bacterial abundance, rather than it being beyond the scope of the study. We recommend that future studies should measure bacterial abundance and if possible composition to better constrain the mesopelagic food web. Please see our response to general comment (3) above.

p. 11, line 22: why not trying to explain this unexpected (and thus interesting) value rather than excluding it from the study?

AC: We provide some possible explanations for this high value in section 3.4, and as noted in a previous comment, choose to do so in this section rather than break the flow of the discussion here which assesses the role of particle associated microbes on POC flux attenuation (review document page 40, line 28)

"Considering the decrease in resident zooplankton populations with depth (Giering et al., 2014), it seems unlikely that FP

production was higher at this depth unless there is a large contribution by diel vertical migrators, and may instead reflect reduced FP loss. However, this scenario could also be due to non steady state conditions with high FP at 500 m reflecting high abundances of FP sinking out from shallower in the water column at a time of previously increased FP production..”

p. 12, line 27: "... which is likely to be less accessible to free-living microbes". This is a  very important statement on which rely the justification that free-living bacterial respiration was not measured here. More details are needed on "less accessible". To what extent?  Some additional quantitative informations supported by adequate references  are  needed.

AC: We have amended these sentences to explain more clearly our justification and to correct our terminology (review document page 47, line 5).

“The direct hydrolysis by attached microbes likely supplies free-living communities with DOC (Cho and Azam, 1988; Karl et al., 1988; Kiørboe and Jackson, 2001). However by definition free-living microbes are not associated with particles and hence do not contribute directly to the loss of large fast sinking POC measured here, and as such we do not consider this loss process. The definition of dissolved and particulate is operational based on the pore size of a GF/F filter, and therefore microbes defined as ‘free-living’ may in fact be able to utilize colloids (Arístegui et al., 2009). However as we only measure the loss of large, fast sinking POC, we exclude free-living bacteria from our analysis of fast sinking POC loss processes. Free-living prokaryotic respiration may account for the ultimate loss of organic carbon from the organic carbon pool but we believe this is reliant on mechanical breakdown of large, fast sinking POC by zooplankton and protozoa (Lampitt et al., 1990; Poulsen and Iversen, 2008; Poulsen et al., 2011) and enzymatic hydrolysis (Smith et al., 1992).”

COMMENTS ON FIGURES

Fig. 1. a: perhaps the authors could reduce the scale of this map and centre it on the PAP site as it is very difficult to see any mesoscale structure at this scale (if this  is what is intended). Also, why not choose a satellite product which encompasses the  sampling period of the study?

AC: We intend this figure to convey the location of the PAP site and hence have chosen a scale that enables the incorporation of identifiable land masses. We choose to plot satellite chlorophyll for 18/06/2015-25/06/2015 (first MSC sample taken on 24/06/2015), in part because particles measured at depth over the first part of the cruise would have originated at the surface during this period. In addition, the cloud cover in the following 8 day satellite image encompasses the PAP study site so is a less useful comparison.

Fig. 3: A log scale on the Y-axis might improve the readability of the different sets of aggregates from each depth. Also separate sinking velocity-size fits for each set of aggregates from distinct depth (one fit by color) could reveal interesting findings, especially if the structure and/or composition of the aggregates are assumed to  vary with depth.

AC: Following your suggestion we have re-plotted the y-axis of figure 3 on a log scale, which also allows us to plot all the outliers. We have investigated relationships of size and sinking velocity for each depth but do not believe we have enough data points within each depth to draw conclusions with certainty. Adding separate fits would, we believe, not be valid and would over complicate the figure.

[revised manuscript text omitted]

---

## Editor Decision (ED1)

Referee #1

The authors have made extensive corrections based on the input of the two reviewers, resulting in an improved manuscript. However, I am still confused by the authors' presentation of their hypothesis and the existing literature; the lack of clarity leaves the entire manuscript still very muddled. This is either due to a continued incorrect reading of the literature on the authors' part, or a lack of clarity in the way they have presented it.

In the abstract: If, following from Giering et al., the authors are contending that two very different carbon budget regimes exist in the "upper" and "lower" mesopelagic ocean, this should be explicitly stated before anything else is presented. A great many studies have shown that respiration and other sinks can exceed carbon inputs in the mesopelagic, but there are very few that have shown the opposite… though this is not what the authors seem to be suggesting: "In particular, it has been suggested that organic-carbon supply exceeds respiration by free-living microbes and zooplankton in the upper mesopelagic." If the authors mean that a very different situation exists in the upper mesopelagic than when considering the mesopelagic as a whole, then this contrast to the existing findings should be explicitly highlighted… and then the authors should very logically proceed to put their study in that context. Since this puts the authors "in the weeds" in terms of their study's broader relevance, they should be very clear about the chain of logic. If this is correct, then the authors should define what they mean by the upper and lower mesopelagic.

Review document p. 42, line 17: I am not sure how the consistent standard errors with depth imply a lack of variability with depth in the factors driving $C_{spec}$. How does this work mathematically? Perhaps I am just missing the authors' point here, but it seems to me one could obtain the same standard error despite large changes in the relative importance of various factors, so long as the increase in the strength of one was accompanied by a precisely complementary decrease in another.

Figure 9: I am not sure that completely removing the error bars was the right decision. Now, it appears the solubilization term has no uncertainty in it! I realize the authors have explained their intent in the caption, but the figure itself is now misleading. Since the authors are attempting to present real data and the results of a thought experiment in the same figure, this has to be very clear. Perhaps the label "Solubilzation" could be changed to "Solubilzation (hypothetical)" or something similar? Also, in (b) where are the error bars on the new zooplankton respiration data? Or the particle respiration data? The errors in these could be combined using statistical methods and presented somehow. Perhaps the entire panel in (b) needs to be marked off as hypothetical?

Referee #2

**Review (2nd round) of "Depth-resolved particle associated microbial respiration in the northeast Atlantic" by Belcher et al.**

**GENERAL COMMENTS AND RECOMMENDATIONS**

I am pleased to notice that the authors have addressed most of my general concerns. The revised version of the manuscript now presents conclusions that are consistent with the hypotheses and measurements conducted. In particular, I appreciate that the main objective of closing the carbon budget in the mesopelagic as noted initially has been replaced by solving imbalances in the upper mesopelagic POC budget.

However, some aspects of the manuscript still need to be revised. Especially, the initial design, objective, methods and results of the roller tank experiment included in this work are still highly questionable and I strongly advise the authors either to thoroughly rework this part of the study or simply remove it from the manuscript.

I am confident that the manuscript will deserve publication in Biogeosciences after the revisions detailed below have been done.

**SPECIFIC COMMENTS**

Note: references made are to the revised manuscript.

Abstract, p. 1, line 19-20: from this sentence it seems that the study is designed to explore an excess of POC supply rather than a missing loss by respiration. The imbalance should be presented the other way around (i.e. the estimated respiration does not balance the observed flux attenuation of POC, suggesting a missing loss).

p. 2, line 7: please add *Steinberg et al.* [2008] to these references.

p. 2, line 9-11: same problem here, please rephrase the other way around.

p. 6, line 24: correct citation is *Logan and Wilkinson* [1990].

p. 6, line 25: again Fractal is not a shape it is a geometry! A spherical particle can have a fractal structure. However a sphere in the Fractal or Euclidean geometries has different structures (but a sphere with a fractal dimension of 3 is equivalent to an Euclidean sphere).

p. 7, line 22: replace *$\mu C$ individual$^{-1}$ h$^{-1}$* by *$\mu g$ C individual$^{-1}$ h$^{-1}$* if it is what was intended.

p. 8, line 28: please change *"The low $R^2$ shows that there is variability around this relationship, suggesting some heterogeneity in PA composition"*, to *"The low $R^2$ suggests that the influence of particle size on the sinking velocity is limited and that particle composition may exert a higher influence".*

**ADDITIONAL COMMENTS TO AUTHOR'S RESPONSES**

GENERAL COMMENTS AND RECOMMENDATIONS

Note: references made are to the initial manuscript and first round of review

(3) *"However, during the cruise a microbiology team performed parallel MSC deployments devoted to linking aggregate carbon content with microbial abundance."*
How long will it take for these data to be produced? It might be worthy to wait for these and include it in the present manuscript.

SPECIFIC COMMENTS

p. 4, line 25: *"Classifications were done manually by A. Belcher based on particle appearance. The morphologies were distinct allowing confident classification."*
A manual classification should usually be avoided because highly subjective to the operator. You need to detail what criteria were used to decide how to sort the particles. I also find really surprising that natural particles had morphologies distinct enough so that they can be classified so easily by hand. Can you provide some kind of evidence that no mixed shape particles were observed?

p. 6, line 1: *"... we observed particle formation after two days and aggregates increased in size during the incubation period."*
Why then did you write in the initial manuscript p.10, line 21 *"..., PAr POC contents could be reduced to 2 µg C mm$^{-3}$ over 7 days (time incubated after first signs of aggregate formation)"*? The first signs of aggregation were obtained after 2 or 7 days?!
*"Additionally, a large number of the aggregates formed in the study of Iversen and Ploug (2010) fall in the range of sinking velocities that we measure on our roller tank formed aggregates (50-150 m d$^{-1}$), suggesting that aggregation processes are not inhibited at this speed."*
Again, there is absolutely no point in comparing the sinking velocities of particles made in different roller tank experiments from different primary particles and measured at different times. The only way to identify a potential effect of tank rotation speed on aggregation kinetic would be by comparing the time of apparition of the first aggregates from two roller tank experiments using the same material incubated at the same concentrations (preferably two identical phytoplankton cultures at the same stage), but using two different rotation speeds.

p. 7 lines 16-17: be careful with the use of "significant relationship" and "significant correlation". The low *p-value* indicates that the result of the statistic test is significant. However, the low $R^2$ suggests an absence of correlation between PA sinking velocity and ESD. This is probably what you call "variability around the relationship".

p. 10, line 15: *"... many studies still utilize roller tank to collect particles. We therefore thought that we could make additional useful comparisons of respiration rates".*

Certainly not! Roller tank experiment are only designed to form particles artificially, not to collect them. The way you justify why you conducted this roller tank experiment is still not satisfying. Why did you want to compare respiration rates of roller-tank made particles with natural particles? (especially because you noted that these particles were not sampled at the same depth).

*"...in an attempt to assess the aggregation potential in the most productive water strata"*

How do you assess the 'aggregation potential'? Roller tank experiments are unfortunately useless at such a task, because they cannot be used for quantitative studies [*Jackson*, 1994].

Based on this, I am still not convinced of the interest of including this roller tank experiment in the manuscript and will let the editor decides whether it has to be removed or not.

REFERENCES CITED

Jackson, G. A. (1994), Particle trajectories in a rotating cylinder: implications for aggregation incubations, *Deep-Sea Research Part I: Oceanographic Research Papers*, *41*(3), 429--437, doi:10.1016/0967-0637(94)90089-2.

Logan, B. E., and D. B. Wilkinson (1990), Fractal geometry of marine snow and other biological aggregates, *Limnology and Oceanography*, *35*(1), 130-136, doi:10.4319/lo.1990.35.1.0130

Steinberg, D. K., B. A. V. Mooy, K. O. Buesseler, P. W. Boyd, T. Kobari, and D. M. Karl (2008), Bacterial vs. zooplankton control of sinking particle flux in the ocean's twilight zone, *Limnology and Oceanography*, *53*(4), 1327.

---

## Author Response (AR2)

**"Depth-resolved particle associated microbial respiration in the northeast Atlantic" *by* A. Belcher et al.**

AC: We would like to thank both reviewers for taking the time to look at our manuscript. We have addressed below the specific comments you have made. In text references refer to the revised manuscript document. In addition, we have edited our text in section 4.4, making an estimate of the rate of respiration on slow-sinking particles to support our hypothesis that fragmentation is an important loss term to the fast-sinking carbon pool (page 15, line 23-33).

"We hypothesize, in line with a growing number of other studies (Cavan et al., 2015; Collins et al., 2015; Iversen et al., 2010), that zooplankton living in the upper mesopelagic may stimulate the loss of large, fast-sinking POC via fragmentation from sloppy feeding, swimming activities and/or microbial gardening (Iversen and Poulsen, 2007; Mayor et al., 2014). Fast-sinking particles can reach the deep ocean with minimal degradation due to the short time in which they are available for degradation (Iversen and Ploug, 2010). Conversely, once fragmented, the increased residence times (in terms of their sinking velocity) of slow and non-sinking POC allows removal to occur at low rates, on longer timescales by microbial respiration. As yet, we are not aware of any published studies measuring respiration rates on slow-sinking POC. Assuming that all of our measured excess POC (i.e. not explained by particle associated microbial respiration, zooplankton respiration or solubilization) in the upper 36-200 m is turned into slow-sinking POC with an average sinking velocity of 9 m d$^{-1}$ (Alonso-González et al., 2010; Riley et al., 2012), we can calculate the respiration rate required to completely remove this excess POC based on equation 5. We estimate that this slow-sinking material would need to be respired at a rate of 0.08 d$^{-1}$ (0.04-0.13 d$^{-1}$), which is not too dissimilar to the rates we have measured on fast-sinking particles, thus providing support to the hypothesis of fragmentation."

**Referee #1**

The authors have made extensive corrections based on the input of the two reviewers, resulting in an improved manuscript. However, I am still confused by the authors' presentation of their hypothesis and the existing literature; the lack of clarity leaves the entire manuscript still very muddled. This is either due to a continued incorrect reading of the literature on the authors' part, or a lack of clarity in the way they have presented it.

AC: Thank you for your comments, we think that the input from both reviewers allowed us to make some good improvements in the last round of the review. We have addressed your comments from the second round of review below, and in particular have reworded sections in the introduction and discussion to ensure the clarity of our work with respect to existing literature.

In the abstract: If, following from Giering et al., the authors are contending that two very different carbon budget regimes exist in the "upper" and "lower" mesopelagic ocean, this should be explicitly stated before anything else is presented. A great many studies have shown that respiration and other sinks can exceed carbon inputs in the mesopelagic, but there are very few that have shown the opposite… though this is not what the authors seem to be suggesting: "In particular, it has been suggested that organic-carbon supply exceeds respiration by free-living microbes and zooplankton in the upper mesopelagic." If the authors mean that a very different situation exists in the upper mesopelagic than when considering the mesopelagic as a whole, then this contrast to the existing findings should be explicitly highlighted… and then the authors should very logically proceed to put their study in that context. Since this puts the authors "in the weeds" in terms of their study's broader relevance, they should be very clear about the chain of logic. If this is correct, then the authors should define what they mean by the upper and lower mesopelagic.

AC: We apologise for any lack of clarity in describing previous findings of carbon imbalances in the mesopelagic. You are correct in that we are addressing here the depth resolved imbalances that have recently been found by Giering et al. 2014. We are not aware of any other studies that have taken such a depth resolved approach, and most take a static upper mesopelagic boundary of 100-200 m. By taking the mixed layer depth as the upper boundary of the mesopelagic layer, Giering et al. 2014 were able to better capture the region of most rapid POC attenuation and found an oversupply of POC (compared to heterotrophic respiration) in the top 50-150m. We believe that the use of a dynamic boundary such as the mixed layer depth, or euphotic zone (as has been recommended by Buesseler and Boyd 2009) would reveal this oversupply in other regions.

In order to improve clarity we have tightened up our use of 'upper mesopelagic', and define it in section 3.4 (page 9, lines 2-4):

"Based on our sampling depths, we define the upper mesopelagic (36-128 m) as the region where the most rapid POC flux attenuation occurs, and the mid mesopelagic as the region below (128-500 m) where we observe a slower decrease and possibly even an increase in POC flux below 128 m."

We have made the following changes in the introduction, (page 2 lines 16-24):

"A recent study managed to close the mesopelagic carbon budget between 50-1000 m in the North Atlantic (Giering et al., 2014) by making key changes to the terms included in the budget. However, Giering et al. (2014) found large and compensating imbalances between sources and sinks in upper and lower mesopelagic layers, with an excess of POC supply to the upper mesopelagic (50-150 m depth), and an excess of respiration in the lower mesopelagic (150-1000 m depth). To the best of our knowledge, this oversupply in the upper mesopelagic has not previously been identified, likely because previous budget studies have not taken a depth resolved approach. In addition, most previous studies use a fixed upper mesopelagic boundary of 100-200 m, rather than a dynamic upper boundary (such as base of the mixed layer, Buesseler and

Boyd, 2009) and therefore may have missed the region of most rapid POC attenuation. Our understanding of the mesopelagic carbon budget is therefore still incomplete."

Additionally we have edited section 4.3 (page 12, lines 26-33):

5   "A number of previous studies have revealed large imbalances in the mesopelagic carbon budget with heterotrophic organic carbon demand (typically assessed from 100 to 1000 m) exceeding POC supply by 2-3 orders of magnitude, (Baltar et al., 2009; Burd et al., 2010; Reinthaler et al., 2006; Steinberg et al., 2008). Recently advances in our understanding enabled the mesopelagic carbon budget at the PAP site to be balanced over 50-1000 m (Giering et al., 2014). However, a more in depth analysis revealed an imbalance when the upper (50-150 m) and lower mesopelagic (150-1000 m) were examined separately,

10  with an oversupply of POC in the upper mesopelagic and an undersupply in the lower mesopelagic (Giering et al., 2014). We find the same imbalance in the upper mesopelagic and as particle associated respiration was not directly measured in the aforementioned study, we assess whether this term could help to explain observed imbalances."

Review document p. 42, line 17: I am not sure how the consistent standard errors with depth imply a lack of variability with

15  depth in the factors driving C_spec. How does this work mathematically? Perhaps I am just missing the authors' point here, but it seems to me one could obtain the same standard error despite large changes in the relative importance of various factors, so long as the increase in the strength of one was accompanied by a precisely complementary decrease in another.

AC: Thank you for pointing this out, we have not been clear with our intended message and agree that counteracting factors could also lead to consistency in Cspec with depth. We have amended the sentences as follows to clarify (page 10 line 22-

20  25):

"There was large variability in $C_{spec}$ for individual aggregates within each depth horizon (full dataset range: 0.002-0.031 d$^{-1}$, Fig, 7), but average values of $C_{spec}$ were quite constant with depth. This suggests that the factors driving the variability in $C_{spec}$ are either also quite constant with depth or counteracting."

25  Figure 9: I am not sure that completely removing the error bars was the right decision. Now, it appears the solubilization term has no uncertainty in it! I realize the authors have explained their intent in the caption, but the figure itself is now misleading. Since the authors are attempting to present real data and the results of a thought experiment in the same figure, this has to be very clear. Perhaps the label "Solubilzation" could be changed to "Solubilzation (hypothetical)" or something similar? Also, in (b) where are the error bars on the new zooplankton respiration data? Or the particle respiration data? The

30  errors in these could be combined using statistical methods and presented somehow. Perhaps the entire panel in (b) needs to be marked off as hypothetical?

AC: Thank you for your suggestions as to how to deal with the unconstrained estimates of solubilization. We have revisited the literature in an attempt to better constrain our error assessment of soluilization. We have adjusted our range of

solubilization estimates based on the modelling study of Anderson and Tang (2010) and study of Grossart and Ploug (2001). This is described in the text (page 13, lines 8-20):

"This solubilization to DOC is likely to fuel the respiration of free-living microbes. Smith et al. (1992) estimated that 97% of the hydrolysates produced by bacteria in marine snow were released, with the remaining 3% being utilized by bacteria in the aggregate. However, this value was based on nitrogen-rich amino acids in fresh aggregates (from the upper 25 m) and hydrolysis for carbon is likely lower as it is lost more slowly than nitrogen from sinking particles. Additionally, solubilization losses are probably lower in older detritus (Anderson and Tang, 2010); Grossart and Ploug (2001) estimated that 26% of the POC was taken up by particle associated bacteria on old aggregates. To calculate potential hydrolysis of carbon from particles, we follow Anderson and Tang (2010) and conservatively assume a value of 50% (i.e. assuming our measured loss via respiration is 50% of the total POC loss via particle associated microbes). This value sits between Smith et al.'s (1992) value and carbon solubilization losses of <30% measured in copepod fecal pellets which are much less porous (Møller et al., 2003). We conservatively set upper and lower bounds of 30 and 80% solubilization based on the aforementioned studies. With additional loss of fast-sinking POC via solubilization we find particle associated microbes can explain 25% (9-72% based on Monte Carlo analysis on respiration rates and the above range in solubilization) of POC losses over the upper 36-200 m (Fig. 9b)."

We have amended the figure and figure legend as below to reflect these changes:

[Figure]

"Figure 9: Balance of processes controlling fast-sinking POC flux attenuation. (a) Comparison of observed POC loss (black bars) and estimated POC loss based on particle associated microbial respiration (grey bars) over two depth horizons (36-128 m, and 128-500 m). (b) POC sources and sinks in the upper 200m. Additional losses ('sinks') via solubilization of POC to DOC by particle associated microbes (estimate based on respiration, see section 4.3), and zooplankton respiration (estimate based on zooplankton biomass and allometric equations). Error bars show uncertainties from sensitivity analysis (see text for details)."

**Referee #2**

Review (2nd round) of "Depth-resolved particle associated microbial respiration in the northeast Atlantic" by Belcher et al.

**GENERAL COMMENTS AND RECOMMENDATIONS**

5   I am pleased to notice that the authors have addressed most of my general concerns. The revised version of the manuscript now presents conclusions that are consistent with the hypotheses and measurements conducted. In particular, I appreciate that the main objective of closing the carbon budget in the mesopelagic as noted initially has been replaced by solving imbalances in the upper mesopelagic POC budget. However, some aspects of the manuscript still need to be revised. Especially, the initial design, objective, methods and results of the roller tank experiment included in this work are still

10   highly questionable and I strongly advise the authors either to thoroughly rework this part of the study or simply remove it from the manuscript. I am confident that the manuscript will deserve publication in Biogeosciences after the revisions detailed below have been done.

AC: Thank you for taking the time to review our manuscript for a second time. We have addressed your further comments and suggestions below. In particular, we have taken your advice and have removed the roller tank work from the manuscript.

15   Considering the uncertainties and limitations of the roller tank data, we agree that they are not strong enough to draw useful conclusions from and do not add anything to the manuscript (see additional comments below). We think removing them helps make the manuscript more focussed and does not affect the conclusions we have made.

**SPECIFIC COMMENTS**

20   Note: references made are to the revised manuscript.

Abstract, p. 1, line 19-20: from this sentence it seems that the study is designed to explore an excess of POC supply rather than a missing loss by respiration. The imbalance should be presented the other way around (i.e. the estimated respiration does not balance the observed flux attenuation of POC, suggesting a missing loss).

25   AC: Amended as follows to clarify our intent:

"In particular, it has been suggested that respiration by free-living microbes and zooplankton in the upper mesopelagic are too low to balance observed flux attenuation of POC within this layer."

p. 2, line 7: please add Steinberg et al. [2008] to these references.

30   AC: Added reference

p. 2, line 9-11: same problem here, please rephrase the other way around.

AC: Rephrased:

"Ultimately carbon is lost from the organic carbon pool as dissolved inorganic carbon via respiration, and hence, in theory at steady state, community respiration should be balanced by the loss (attenuation) of POC (Burd et al., 2010)."

p. 6, line 24: correct citation is Logan and Wilkinson [1990].

AC: We refer to the Iversen and Ploug (2010) paper as an example of a study where a power law relationship was found between particle ESD and sinking velocity. We have added in the Logan and Wilkinson (1990) reference to our statement of divergence of marine snow aggregates from Stoke's Law (page 6, lines 25-27).

"The choice of a power law relationship, based on the findings of previous studies (e.g. Iversen and Ploug, 2010) was motivated by the observed divergence of marine snow aggregates from Stoke's Law due to their irregular shapes (Logan and Wilkinson, 1990)."

p. 6, line 25: again Fractal is not a shape it is a geometry! A spherical particle can have a fractal structure. However a sphere in the Fractal or Euclidean geometries has different structures (but a sphere with a fractal dimension of 3 is equivalent to an Euclidean sphere).

AC: Thank you for highlighting our mistake in terminology. We have amended the sentence (shown above), to rectify this.

p. 7, line 22: replace $\mu C$ individual$^{-1}$ h$^{-1}$ by $\mu g$ C individual$^{-1}$ h$^{-1}$ if it is what was intended.

AC: Amended, thank you for spotting our mistake.

p. 8, line 28: please change "The low $R^2$ shows that there is variability around this relationship, suggesting some heterogeneity in PA composition", to "The low $R^2$ suggests that the influence of particle size on the sinking velocity is limited and that particle composition may exert a higher influence".

AC: Changed as recommended.

**ADDITIONAL COMMENTS TO AUTHOR'S RESPONSES**

GENERAL COMMENTS AND RECOMMENDATIONS

Note: references made are to the initial manuscript and first round of review

(3) "However, during the cruise a microbiology team performed parallel MSC deployments devoted to linking aggregate carbon content with microbial abundance."

How long will it take for these data to be produced? It might be worthy to wait for these and include it in the present manuscript.

AC: We have spoken to the scientists working on these data, and unfortunately the measurements require a much greater time to complete analysis and be available for use (>6 months). We think that the dataset we present in our manuscript will

be useful to the scientific community without ancillary microbiology data and are keen to make our data available for others to use. In the future when the full microbiology data becomes available, publications by their team will provide further insight into the particle associated microbial community making further scientific advancements.

5   SPECIFIC COMMENTS

p. 4, line 25: "Classifications were done manually by A. Belcher based on particle appearance. The morphologies were distinct allowing confident classification."
A manual classification should usually be avoided because highly subjective to the operator. You need to detail what criteria
10  were used to decide how to sort the particles. I also find really surprising that natural particles had morphologies distinct enough so that they can be classified so easily by hand. Can you provide some kind of evidence that no mixed shape particles were observed?
AC: Classification of particles involved a number of steps. The particles were first categorised by type, splitting into fecal pellets, marine snow aggregates and other detritus. We then calculate the area and volume of marine snow aggregates using
15  the geometric equations for a sphere or ellipse in line with previous studies (e.g. Ploug et al. 2008; Iversen and Ploug 2013; Ebersbach and Trull 2008), based on the fact that most marine aggregates observed in situ are of these geometries (Alldredge 1998 and refs within). We believe that any small errors in the exact 2D particle shape are greatly outweighed by the conversion from a 2D area to a 3D volume (a method used by previous studies, e.g. Ebersbach and Trull, 2008). However, these area conversions were only carried out to assess % composition in each category (Figure 2) and do not affect our
20  respiration measurements. We were able to measure full X, Y and Z dimensions for particles used for respiration measurements whilst they were suspended in the flow chamber.

p. 6, line 1: "... we observed particle formation after two days and aggregates increased in size during the incubation period."
Why then did you write in the initial manuscript p.10, line 21 "..., PAr POC contents could be reduced to 2 µg C mm-3 over
25  7 days (time incubated after first signs of aggregate formation)"? The first signs of aggregation were obtained after 2 or 7 days?!
AC: The first signs of aggregation were after 2 days, and hence we calculate the loss of POC that may have occurred from these aggregates from the time of formation (day 2) to the time of measurement (day 9), which is a period of 7 days. As we have now taken out the roller tank analysis, this sentence has in any case been removed.

"Additionally, a large number of the aggregates formed in the study of Iversen and Ploug (2010) fall in the range of sinking velocities that we measure on our roller tank formed aggregates (50-150 m d-1), suggesting that aggregation processes are not inhibited at this speed."
Again, there is absolutely no point in comparing the sinking velocities of particles made in different roller tank experiments

from different primary particles and measured at different times. The only way to identify a potential effect of tank rotation speed on aggregation kinetic would be by comparing the time of apparition of the first aggregates from two roller tank experiments using the same material incubated at the same concentrations (preferably two identical phytoplankton cultures at the same stage), but using two different rotation speeds.

5 AC: We appreciate your comments and knowledge of roller tank studies. Taking due consideration of the latter and to the uncertainties in the data, in part due to the long incubation time, and because we do not have a marine snow catcher sample at the same depth to compare to, we have removed this element from the manuscript. This does not influence our conclusions as to the role of particle associated microbial respiration in the attenuation of POC.

10 p. 7 lines 16-17: be careful with the use of "significant relationship" and "significant correlation". The low p-value indicates that the result of the statistic test is significant. However, the low R2 suggests an absence of correlation between PA sinking velocity and ESD. This is probably what you call "variability around the relationship".

AC: We have amended the sentence with 'variability around the relationship' as suggested in the specific comment –'p. 8, line 28' – above. We only use 'significant' when a statistical test has been carried out and the calculated $p$ value shows the
15 result to be significant.

p. 10, line 15: "... many studies still utilize roller tank to collect particles. We therefore thought that we could make additional useful comparisons of respiration rates".

Certainly not! Roller tank experiment are only designed to form particles artificially, not to collect them. The way you justify
20 why you conducted this roller tank experiment is still not satisfying. Why did you want to compare respiration rates of roller-tank made particles with natural particles? (especially because you noted that these particles were not sampled at the same depth).

"...in an attempt to assess the aggregation potential in the most productive water strata"

How do you assess the 'aggregation potential'? Roller tank experiments are unfortunately useless at such a task, because they
25 cannot be used for quantitative studies [Jackson, 1994].

Based on this, I am still not convinced of the interest of including this roller tank experiment in the manuscript and will let the editor decides whether it has to be removed or not.

AC: Roller tank data and discussion of these data have been removed – see comment above.

[revised manuscript text omitted]

---

## Author Response (AR3)

**"Depth-resolved particle associated microbial respiration in the northeast Atlantic" *by* A. Belcher et al.**

**Author Response**

**Reviewer 1:**

The authors' revisions during the second round of review have improved the manuscript dramatically. Their message is clear, and their presentation now does a fantastic dataset adequate justice. I have only a two very minor (easily fixable) comments pertaining to the way the authors have phrased their statements of the carbon mass imbalance in response to the second reviewer's comments.

AC: Thank you again for taking the time to read our revised manuscript and for your comments and advice throughout the review process. Thank you for noting the error in our phrasing regarding the carbon mass imbalance, we have amended these as described below and as shown in the track changed version.

p. 9: Currently: "In particular, it has been suggested that respiration by free-living microbes and zooplankton in the upper mesopelagic are too low to balance observed flux attenuation of POC within this layer." As in my comment below, this phrasing doesn't make sense mathematically. Respiration and flux attenuation are both loss processes (sink terms), so there is no source here that would provide any sort of balance. I understand what the authors are attempting to say, but a small change should be made to make this correct, in a mass balance sense. Could try "... upper mesopelagic is too low to explain observed flux attenuation of POC within this layer." or " ... upper mesopelagic is too low to balance observed inputs of POC within this layer." Both would be correct.

AC: Sentence has been corrected to:

"In particular, it has been suggested that respiration by free-living microbes and zooplankton in the upper mesopelagic are too low to explain the observed flux attenuation of POC within this layer."

p. 10: Currently: "Ultimately carbon is lost from the organic carbon pool as carbon via respiration, and hence, in theory at steady state, community respiration should be balanced by the loss (attenuation) of POC (Burd et al., 2010)." I fully understand what the authors are attempting to express here, but it does not make sense mathematically as currently stated. In the mass balance, community respiration isn't balanced by the loss of POC because community respiration is *itself* the loss process. The way it is currently phrased, there is no balancing input term! Instead, how about "... respiration should be balanced by the supply of POC" or something similar. The plain-language description should parallel a written equation. (The authors phrase the situation correctly in the sentence immediately following.)

AC: Sentence has been corrected to:

"Ultimately carbon is lost from the organic carbon pool as dissolved inorganic carbon via respiration, and hence, in theory at steady state, community respiration should be balanced by the supply of POC (Burd et al., 2010)."

**Reviewer 2:**

**GENERAL COMMENTS AND RECOMMENDATIONS**

5   The authors have thoroughly addressed all comments and recommendations made in the 2nd round of review. The manuscript has been considerably improved since the first submission and significantly gained in clarity. In particular, the objectives developed in the introduction, the analyses conducted and the conclusions are now in-line and flow logically along the different sections. By removing all elements related to the roller tank experiment, the authors strengthened the manuscript and enhanced its coherence.

10   I can now strongly recommend this manuscript for publication in Biogeosciences and salute the authors for their rigorous work. Below are some minor technical corrections to be addressed at the authors' convenience.

AC: Thank you for reviewing our manuscript again, we have addressed all your technical corrections below.

**TECHNICAL CORRECTIONS**

15   Note: references made are to the revised manuscript.

p. 2, line 5: add *François et al.* [2002]

AC: Reference added

p. 3, line 5: change "estimated" to "estimate".

AC: Changed

20   p. 4, line 13: "... at which depth the bottles were closed,..." This sentence sounds incorrect. Maybe try to rephrase it.

AC: Changed to "…closed at depth…"

p. 5, line 1: "... and F0 is flux at the reference depth...". Add "the" between "is" and "flux".

AC: Amended

p. 5, line 6: change "aggregations" to "aggregates" if it was the word intended.

25   AC: Changed

p. 5, line 9: "Conversions from PA volume...". It seems like a "to" should follow the "from". Check the sentence.

AC: Sentence changed to:

"Conversions to PA POC from PA volume were based on measurements of POC content of marine aggregates collected at depth (section 2.4)"

30   p. 5, lines 26-27: "PA volumes were calculated from...". This sentence does not read nicely, try to rephrase it.

AC: Sentence changed to:

"The formula of an ellipsoid was used to calculate PA volumes from their x, y, z dimensions, and equivalent spherical diameters (ESD) were computed."

p. 5, line 30: remove the parentheses enclosing "50-200 μm".

AC: Removed

p. 6, line 20: to be consistent you should give the form of the power-law relationship as you did for the exponential fit later in the sentence.

AC: Added:

"of form, $F_z = F_0 (z/z_0)^{-b}$"

p. 6, line 31: "We conduct...". The remaining of the paragraph is at the past tense. I noticed a large number of inconsistencies in the use of present and past tense in the manuscript. Please check this.

AC: We have corrected the tenses in this paragraph and made amendments in the rest of the manuscript where necessary to remove any inconsistencies.

p. 7, line 24: change "(Ikeda, 1985)" to "Ikeda (1985)".

AC: Changed

p. 8, line 2: change "chlorophyll-a" to "chlorophyll *a*".

AC: Changed

p. 8, lines 15 to 18: you use the same kind of explanations for the lack and increase of FP, maybe you could try to combine them.

AC: As we suggest slightly different explanations to explain the changes in FP with depth we choose to keep these two sentences separate for clarity. For example, the lack of FP in the 113 m are unlikely to be due to the DVM of zooplankton as this would tend to increase FP at depth.

p. 8, line 25: "... showing consistency in the composition of the bulk of sinking PA." This is very hard to affirm this considering the multiple controls on aggregate sinking velocity (e.g. composition, structure, size, shape).

AC: We have removed the last part of the sentence as indeed there are many other factors that affect sinking velocity. The sentence now reads:

"Median sinking velocities showed less variability ranging from 11-34 m d$^{-1}$ (10-32 m d$^{-1}$ and 21-62 m d$^{-1}$ for aggregates <0.6 mm (n=74) and >0.6 mm (n=24) ESD respectively)."

p. 8, line 26: again I disagree: there is no significant correlation between size and sinking velocity! And your last sentence (lines 28-29) contradicts this statement. Please fix this.

AC: On line 26 we state that there is no correlation between sinking velocity and depth, i.e. that we do not see an increase or decrease in sinking velocity with depth. We tested both size classes for this relationship but are not referring here to the correlation between size and sinking velocity. We do however find a significant relationship between sinking velocity and ESD (as shown in figure 3, and by the low *p* value), and have swapped the words 'correlated with' with 'related to' to clarify the meaning of our results.

"There was no significant relationship between PA sinking velocity and depth for either size class ($R^2$=0.004, *p*>0.1, n=98). PA sinking velocity was significantly ($R^2$=0.17, *p*<0.0001, n=98) related to ESD (6 outliers, defined as being outside 2

standard deviations from the mean, were excluded in this relationship). The low $R^2$ suggest that the influence of particle size on the sinking velocity is limited and that particle composition may exert a higher influence."

p. 9, line 2: "Consistent with other studies...". Please cite them.

AC: We have added 3 citations as examples of where particle flux is observed to decline rapidly with depth.

"(e.g. Giering et al., 2014; Martin et al., 1987; Riley et al., 2012)."

p. 10, lines 8-9: "This may reflect the heterogeneity of aggregate composition in terms of the availability of labile carbon...". Please rephrase this sentence.

AC: Sentence rephrased to:

"This may reflect the heterogeneity in the availability of labile carbon and/or variation in microbial abundance, composition or activity within the PA."

p. 10, line 25: "quite constant" does not mean anything. Maybe you can replace it by "small variations"?

AC: Changed to

"…but average values of $C_{spec}$ showed only small variations with depth."

p. 12, line 31: "... with an oversupply of POC in the upper mesopelagic and an undersupply in the lower mesopelagic". The "oversupply" and "undersupply" refer to bacterial demand, you should mention it.

AC: Sentences reworded to:

[revised manuscript text omitted]

---

## Author Response (AR4)

**"Depth-resolved particle associated microbial respiration in the northeast Atlantic" *by* A. Belcher et al.**

**Author Response to Editor's report**
**Editor's comments**

5    It is a great pleasure now to accept your manuscript for publication in Biogeosciences. Thank you very much for responding to the reviewers' comments so quickly and carefully.

AC: Thank you for taking the time to check out manuscript and manage the review process. We have made the necessary technical corrections as detailed below and in the track change version.

10    Before it is published, please consider my technical corrections below:

P1, L23 and hereafter: 36-500 m. Please use en dash (–), not hyphen (-) for the range.
AC: Changed to en dash throughout the manuscript
P2, L8: Remove the semicolon (;).
15    AC: Removed
P3, L3: laboratory cultures
AC: Amended
P3, L20: mixed layer depth ≤ 500 m
AC: Changed to "mixed layer depth to 500 m" to explain the depth range we are interested in.
20    P3, L21: the second "poorly" may be replaced by "little" or a similar word.
AC: Changed to "not well"
P5, L10: 0.08 mg C
AC: Amended
P5, L31: Please use the symbol "$\chi$", not alphabet "x".
25    AC: Amended
P6, L22: insert ", where $z*$ is the characteristic remineralization length scale for the flux decrease below $z_0$ as in Buesseler and Boyd, 2009).
AC: Amended
P7, L26: 0.8 mol C (mol O2)^(-1)
30    AC: Amended
P8, L27: "$R^2$" should be italic.
AC: Amended
P9, L29: Table S1
AC: Amended
35    P11, L10: Use "±", not "+/-".
AC: Amended
P12, L29: Remove a space immediately before "carbon".
AC: Removed
P17, L32: the dark ocean's pelagic
40    AC: Amended
P29, L3: Error bars represent ± one standard error of the mean.
AC: Amended

[revised manuscript text omitted]